# A transcriptome atlas of the mouse iris at single-cell resolution defines cell types and the genomic response to pupil dilation

Jie Wang[1,2], Amir Rattner[1], Jeremy Nathans[1,2,3,4]*

[1]Department of Molecular Biology and Genetics, Johns Hopkins University School of Medicine, Baltimore, United States; [2]Howard Hughes Medical Institute, Johns Hopkins University School of Medicine, Baltimore, United States; [3]Department of Neuroscience, Johns Hopkins University School of Medicine, Baltimore, United States; [4]Department of Ophthalmology, Johns Hopkins University School of Medicine, Baltimore, United States

**Abstract** The iris controls the level of retinal illumination by controlling pupil diameter. It is a site of diverse ophthalmologic diseases and it is a potential source of cells for ocular auto-transplantation. The present study provides foundational data on the mouse iris based on single nucleus RNA sequencing. More specifically, this work has (1) defined all of the major cell types in the mouse iris and ciliary body, (2) led to the discovery of two types of iris stromal cells and two types of iris sphincter cells, (3) revealed the differences in cell type-specific transcriptomes in the resting vs. dilated states, and (4) identified and validated antibody and in situ hybridization probes that can be used to visualize the major iris cell types. By immunostaining for specific iris cell types, we have observed and quantified distortions in nuclear morphology associated with iris dilation and clarified the neural crest contribution to the iris by showing that *Wnt1-Cre*-expressing progenitors contribute to nearly all iris cell types, whereas *Sox10-Cre*-expressing progenitors contribute only to stromal cells. This work should be useful as a point of reference for investigations of iris development, disease, and pharmacology, for the isolation and propagation of defined iris cell types, and for iris cell engineering and transplantation.

*For correspondence:
jnathans@jhmi.edu

Competing interest: The authors declare that no competing interests exist.

## Editor's evaluation

Using single nucleus RNA sequencing, the authors have characterized all major cell types in the mouse iris and ciliary body, defined new types of iris stromal and sphincter cells, and shown cell-specific transcriptome responses in the resting, constricted, and dilated states. They have identified and validated antibodies and in situ hybridization probes for visualization of major iris cell types. This work will be a valuable reference for investigations of iris development, disease, and pharmacology.

## Introduction

The iris, a thin disc of light-absorbing tissue with a central opening, is located in front of the lens and divides the eye into anterior and posterior chambers (*Figure 1A*). Muscular contraction within the iris controls the diameter of the central opening – the pupil – to adjust the amount of light impinging on the retina. The iris has been an object of popular and scientific fascination for centuries, most likely as a consequence of its position as the principal target of gaze fixation in interpersonal encounters,

the unconscious responses of the pupil to different psychological states, and the great diversity in iris appearance/pigmentation among humans (*Duke-Elder and Wybar, 1961*; *Larsen and Waters, 2018*).

The iris is also of clinical significance as the site of inflammation in anterior uveitis, as a reporter of neurologic function via the pupillary light response, and as one of the tissues affected in a variety of congenital malformations such as coloboma or aniridia, in which part or all of the iris is missing (*Vaughan et al., 1992*). For more than a century, neural and pharmacologic control of the iris's radial/dilator and circumferential/sphincter muscles has served as a paradigm for the integration of clinical and basic sciences (*Thompson, 1992*; *Williams et al., 2000*; *Wilhelm, 2011*). The superficial location of the iris and its accessibility within an optically clear structure has facilitated quantitative analyses of iris structure, function, and disease, and the effect of genetic variation on iris development and appearance (*Loewenfeld and Newsome, 1971*; *Davis-Silberman and Ashery-Padan, 2008*; *McDougal and Gamlin, 2015*; *Simcoe et al., 2021*).

The human pupil exhibits an ~4 -fold variation in diameter between constricted and dilated states (*Duke-Elder and Wybar, 1961*). Under natural conditions, there are extended periods of pupil constriction, induced by sunlight during the day, alternating with extended periods of pupil dilation, induced by darkness at night. Pharmacologic dilation, produced by topical administration of an alpha-adrenergic agonist and/or a muscarinic antagonist, is widely used in the context of routine ophthalmologic exams and ocular surgery (*Paggiarino et al., 1993*; *Motta et al., 2009*). In response to constriction and dilation, several iris cell types exhibit striking changes in shape and in the arrangement of their plasma membranes and cytoskeletal fibers (*Lim and Webber, 1975a*). At present, there are no known molecular correlates of these cellular responses.

We have undertaken the present study to provide foundational information for future investigations of iris structure, function, development, and disease. Our approach has been to define the transcriptome profiles of all of the major cell types of the mature murine iris by single nucleus (sn)RNAseq, to use those profiles to define the full repertoire of those cell types and the relationships among them, and to explore how the transcriptome changes within each iris cell type in response to pupil dilation or constriction. Using transcription factors as cell type-specific markers, we have also visualized the nuclear deformations that occur in response to pupil dilation and assessed the neural crest origin of different iris cell types.

## Results
### The major cell types of the mouse iris

The diversity of cell types in the iris – which includes muscle, epithelial, and stromal cells – suggested that homogenizing the iris and purifying nuclei might provide a more comprehensive and uniform sampling than would enzymatically dissociating the iris into single cells. Tissue homogenization has the added advantage that regulated RNA synthesis and degradation should largely cease upon cell disruption and dilution of the nuclei into homogenization buffer. By contrast, with enzymatic dissociation preparatory to single-cell (sc)RNAseq, transcriptome changes can be observed during the 30–60 min required for cell dissociation and processing (*Lacar et al., 2016*; *Hrvatin et al., 2018*). A potential disadvantage of snRNAseq relative to scRNAseq is that it is insensitive to changes in RNA abundance arising from changes in RNA stability in the cytoplasm.

To prepare mouse iris and ciliary body samples for snRNAseq, nuclei were purified following: (1) continuous pupil dilation (mydriasis) for six and one-half hours with hourly 5 % phenylephrine +1 % cyclopentolate eye drops (two replicates), (2) continuous pupil constriction (miosis) for six and one-half hours with hourly 5 % pilocarpine eye drops (three replicates), or (3) no treatment (three replicates). Six and one-half hours was chosen as the duration for dilation or constriction to provide sufficient time for any changes in gene expression to be fully manifest. snRNAseq data were obtained from a total of 34,357 nuclei (dilation: 11,027 nuclei; constriction: 8965 nuclei; no treatment: 14,365 nuclei) with a mean of 1760 transcripts detected per nucleus, using the 10 × Genomics Chromium platform (*Supplementary file 1*). The data were analyzed using the Seurat platform. Among independent replicates from mice subject to the same treatment, RNAseq read counts showed pairwise Pearson correlations of 0.98–0.99 (*Figure 1—figure supplement 1*). Therefore, for all subsequent analyses, each set of

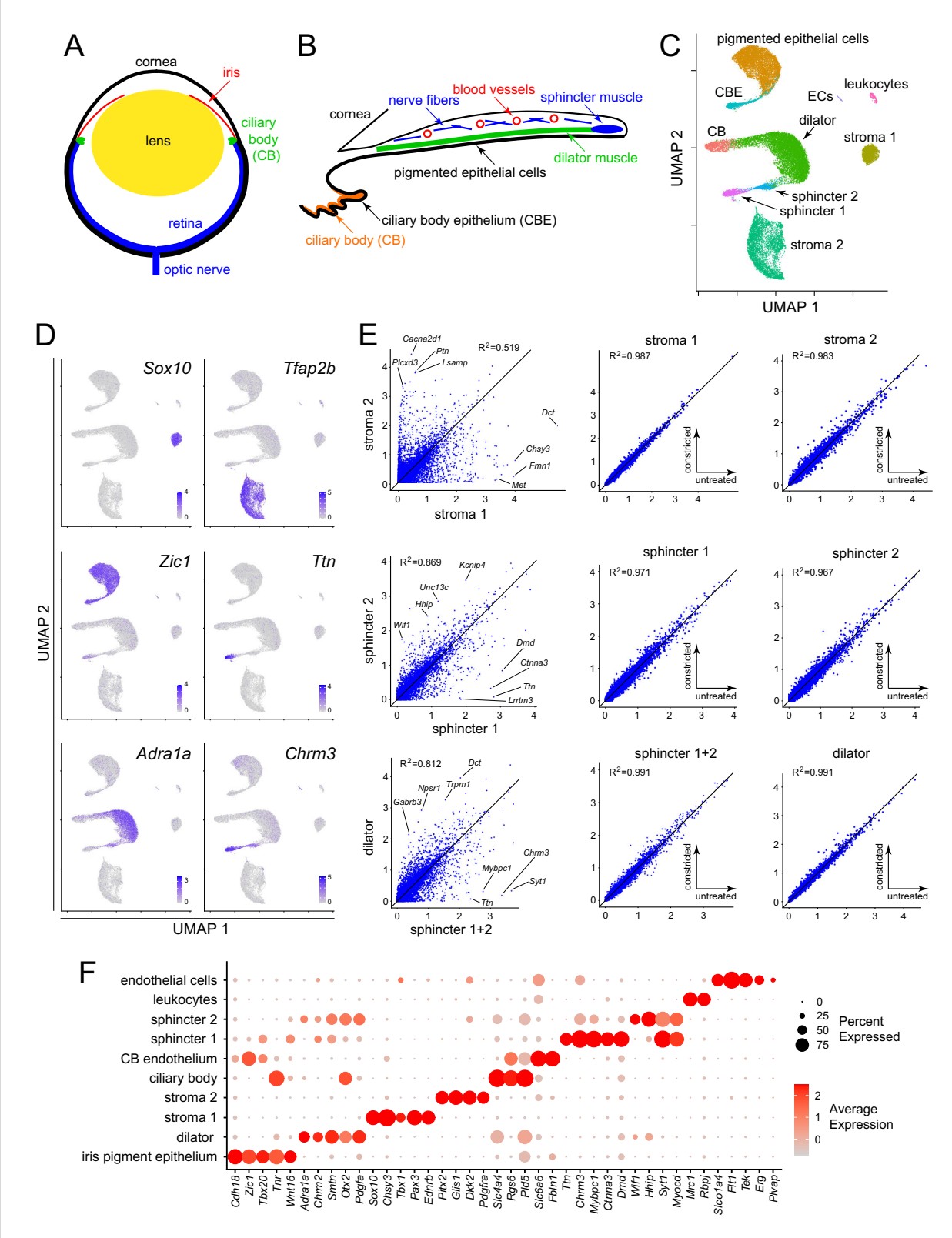

**Figure 1.** Single nucleus RNAseq (snRNAseq) defines the major cell types in the mouse iris. (**A**) Schematic cross section of a mouse eye. (**B**) Schematic cross section of the mouse iris and ciliary body showing the locations of the major structures. (**C**) Uniform Manifold Approximation and Projection (UMAP) representation of snRNAseq from untreated, constricted, and dilated mouse irises, with major cell types indicated. CB, ciliary body; CBE, ciliary body epithelium; ECs, endothelial cells. (**D**) UMAP plots, as in (**C**), with examples of read counts (in blue) for individual transcripts that define

*Figure 1 continued on next page*

*Figure 1 continued*

distinct iris cell types. Color intensities represent the scaled average expression level per cell. (**E**) Scatter plots of snRNAseq read counts ($\log_{10}$ average normalized expression) and $R^2$ values for pairwise comparisons of all transcripts between the indicated cell types. For each of the three rows, the left plot is the comparison of interest, obtained from untreated irises, and the right two plots show comparisons to assess technical variation between different snRNAseq datasets (untreated vs. constricted irises). (**F**) Dot plot showing transcript abundances across iris cell types for a subset of transcripts that define distinct cell types. In this and all other dot plots, the percentage of cells expressing the indicated transcript is represented by the size of the circle, and the average expression level per cell (the scaled average SCTransform normalized expression calculated with Seurat) is represented by the intensity of the circle.

The online version of this article includes the following figure supplement(s) for figure 1:

**Figure supplement 1.** Pairwise Pearson correlations for all of the iris single nucleus (sn)RNAseq libraries.

**Figure supplement 2.** Uniform Manifold Approximation and Projection (UMAP) plots for all of the iris single nucleus (sn)RNAseq libraries.

replicate samples were merged into a single dataset. Female albino mice were used throughout this study to facilitate visualization of fluorescent antibody and in situ hybridization (ISH) probes.

Ten iris cell clusters were identified with Seurat, and their identities were subsequently assigned by immunostaining and ISH, as described below. A Uniform Manifold Approximation and Projection (UMAP) plot of the snRNAseq data revealed a distinctive distribution of the 10 clusters (*Figure 1B and C*): (1) a comma shape, consisting of the iris pigment epithelium (PE) and the adjacent and contiguous ciliary body epithelium (CBE), (2) a C-shaped arc in which dilator muscle connects to non-epithelial ciliary body cells on one side and sphincter muscle on the other side, (3) two widely separated clusters of stromal cells (referred to as stroma 1 and stroma 2), and (4) small clusters of leukocytes and endothelial cells. UMAP plots are well matched across treatment conditions and independent replicates (*Figure 1—figure supplement 2*), with the only visually apparent difference being a small shift in the location of the dilator muscle cell cluster upon pupil dilation.

Some cluster identities could be inferred based on existing physiologic data. For example, classic pharmacologic studies have shown that the dilator muscle is responsive to alpha-adrenergic agonists and the sphincter muscle is responsive to muscarinic agonists. As shown in *Figure 1D*, transcripts coding for the alpha-1A adrenergic receptor (*Adra1a*) are enriched in the dilator cluster, and transcripts coding for muscarinic acetylcholine receptor 3 (*Chrm3*) are enriched in the sphincter cluster. Interestingly, the sub-cluster of sphincter cells closest to the dilator cluster (referred to as the 'sphincter 2' cluster) contains both transcripts. Other transcripts that show high cluster specificity include transcription factors *Sox10* (stroma 1), *Tfap2b* (stroma 2), and *Zic1* (epithelial cells) (*Figure 1D*). Transcripts coding for the gigantic muscle protein titin (*Ttn*) are restricted to the most distal sub-cluster of sphincter cells (referred to as the 'sphincter 1' cluster).

The distinctions between clusters rest on differences in the expression of multiple genes, a small fraction of which are shown in *Figure 1F* (see also *Supplementary file 2*). That the stroma 1 vs. stroma 2 and the sphincter 1 vs. sphincter 2 distinctions reflect transcriptome differences substantially larger than the differences associated with sample-to-sample variability is apparent in comparing scatter plots of snRNAseq read counts for these pairwise comparisons (from untreated irises) to scatter plots for each cluster from untreated vs. constricted irises (*Figure 1E*). [There is almost no effect of pharmacologic constriction on iris gene expression, as described more fully below.] *Figure 1E* also shows a scatter plot comparison of dilator vs. sphincter muscles (i.e., sphincter 1 and sphincter 2 clusters combined), showing that the scatter in the cross-cluster comparison is far greater than for the intra-cluster comparisons. These data indicate that the mouse iris contains two distinct types of stromal cells and three distinct types of smooth muscle cells.

## Differences between dilator and sphincter muscles

A subset of markers that distinguish the different classes of iris smooth muscle cells were investigated by immunostaining (*Figure 2A*). Smooth muscle actin (SMA, the product of the *Acta2* gene) is expressed in all three classes of smooth muscle cells (*Figure 2C, D and F–H*). In iris cross sections, the strands of dilator muscle typically weave in and out of the plane of section (*Figure 2B–G*), as inferred from SMA immunostaining of iris flat mounts (*Figure 2H*). Multiple markers for sphincter 1 cells – including neuronal pentraxin-2 (*Nptx2*), NeuN (*Rbfox3*), catenin-alpha3 (*Ctnna3*), muscle creatine kinase (*Ckm*), and titin (*Ttn*) – localized throughout the pupil-proximal 10–15% of the iris (*Figure 2B–H*). This region coincides with the region of circumferential muscle fibers visualized by

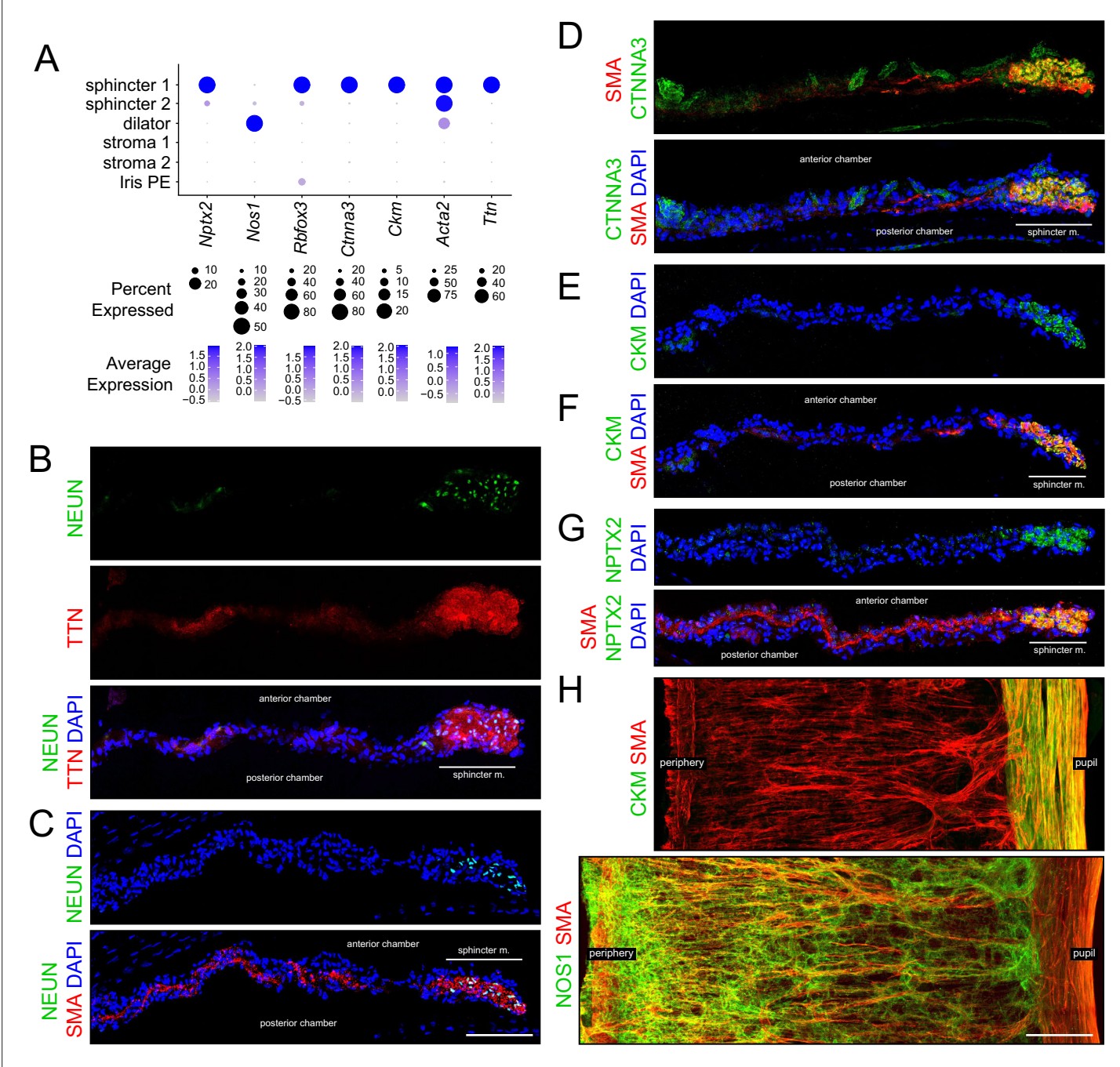

**Figure 2.** Immunostaining for markers that distinguish sphincter and dilator muscles. (**A**) Dot plot (as described in *Figure 1F*) showing transcript abundances across a subset of iris cell types for transcripts coding for the markers shown in (**B–H**). *Rbfox3* codes for NEUN. Scales are individualized for each transcript to accommodate the large differences in their abundances. (**B–G**) Cross sections of untreated iris immunostained as indicated. In this and all other figures showing iris cross sections, the cornea is above and the lens is below the region shown. The pupil is to the right and the region encompassing the sphincter muscle is indicated. (**H**) Flat mounts of untreated iris immunostained as indicated. The sphincter muscle occupies the right-most 15–20% of the iris. For each immunostaining analysis in this and subsequent figures, iris cross sections were stained from at least five mice and iris whole mounts were stained from at least two mice. All images are at the same magnification. Scale bars in (**C**) and (**H**), 100 μm.

The online version of this article includes the following figure supplement(s) for figure 2:

**Figure supplement 1.** Abundances of transcripts coding for signal transduction components in each of the principal iris cell types.

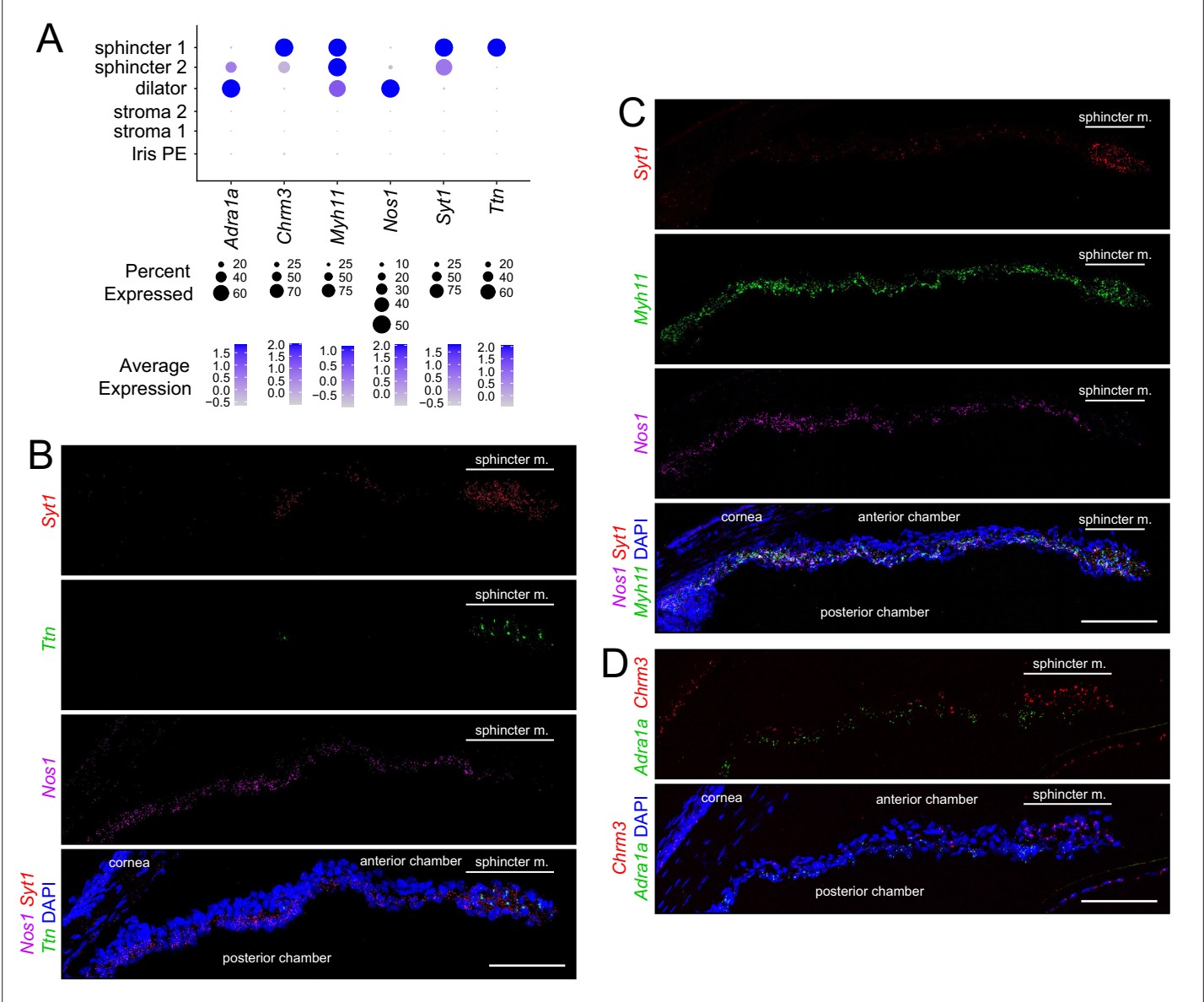

**Figure 3.** In situ hybridization for markers that distinguish sphincter and dilator muscles. (**A**) Dot plot (as described in *Figure 2A*) showing transcript abundances across a subset of iris cell types for transcripts coding for markers shown in (**B–D**). (**B–D**) Cross sections of untreated iris hybridized with the indicated probes. All images are at the same magnification. For each in situ hybridization analysis in this and subsequent figures, iris cross sections were hybridized from at least three mice. Scale bars, 100 μm.

SMA immunostaining, implying that sphincter 1 and sphincter 2 cells are intermingled with each other within the mass of sphincter muscle fibers. Nitric oxide synthase-1 (*Nos1*), the NOS isoform enriched in the nervous system and skeletal muscle, is specifically expressed in dilator muscle and localizes in a pattern reciprocal to that of CKM (*Figure 2H*).

Markers for iris smooth muscle cells were also localized by ISH, using the RNAscope platform (*Figure 3A*). This analysis localized transcripts for *Nos1* to dilator muscle, *Ttn* to sphincter muscle, and myosin heavy chain-11 (*Myh11*) to both dilator and sphincter muscles (*Figure 3B and C*). Transcripts coding for synaptotagmin-1 (*Syt1*), a calcium sensor that regulates neurotransmitter release, localized to sphincter 1 and sphincter 2 cells (*Figure 3B and C*). As predicted from the UMAP analysis, *Adra1a* transcripts are enriched in dilator muscle and *Chrm3* transcripts are enriched in sphincter muscle (*Figure 3D*).

Except for the adrenergic and cholinergic receptors, the physiologic significance of differential gene expression in the different iris smooth muscle subtypes is unclear. Titin, the largest member of the mammalian proteome (~27,000 to ~35,000 amino acids, depending on the splice isoform) connects the Z-disk to the M-band in striated muscle and appears to act as an elastic element, responding to muscle tension by progressively unfolding its many small domains (*Chauveau et al., 2014*). A similar elastic role in the sphincter 1 cells appears plausible in light of the 4-fold change in pupil diameter – and, hence, sphincter muscle length – with dilation and constriction. Why Titin would not similarly be required in dilator muscle is unclear. The localization of CTNNA3 to sphincter 1 cells represents a second structural/cytoskeletal distinction between sphincter and dilator muscle. Alpha-catenins heterodimerize with beta-catenins at adherins junctions and they regulate actin filament assembly, coordinating junctional and cytoskeletal architectures (*Takeichi, 2018*).

The localization to sphincter cells of NeuN, neuronal activity-regulated pentraxin-2 (NPTX2), and synaptotagmin-1 (SYT1) – all of which are enriched in the nervous system – suggests that gene expression in sphincter cells has a partially neuronal character. NeuN/RBFOX3 is an RNA-binding protein that regulates differential pre-mRNA splicing (*Kim et al., 2009*; *Kim et al., 2013*; *Duan et al., 2016*). NPTX2 (formerly NARP), the product of a neuronal immediate-early gene, is an extracellular protein that enhances neurite outgrowth, clusters AMPA receptors, and promotes synaptic plasticity (*Tsui et al., 1996*; *O'Brien et al., 1999*). SYT1 promotes calcium-triggered synaptic vesicle fusion by arresting the SNARE complex in the pre-fusion state under low calcium conditions and then allowing the SNARE complex to 'zipper up' its alpha helices and promote membrane fusion in high calcium (*Zhou et al., 2017*; *Ramakrishnan et al., 2020*).

The cell-type-specific transcriptomes derived from snRNAseq provide an opportunity to systematically assess signal transduction components that regulate iris smooth muscle function. In amphibia, birds, fish, and non-primate mammals, pupil diameter is controlled both by a retina-brainstem-sympathetic/parasympathetic-iris smooth muscle circuit and by the intrinsic light sensitivity of the iris musculature (*Douglas, 2018*). Studies of knockout mice show that the intrinsic pathway is mediated by the G-protein coupled receptor (GPCR) photopigment melanopsin (OPN4) via activation of phospholipase C-beta2 and beta4 (PLCB2 and PLCB4) and calcium release by inositol 1,4,5-trisphosphate receptor 1 (IP3R1) (*Xue et al., 2011*; *Wang et al., 2017*). Our transcriptome analysis shows that of all the genes coding for rhodopsin-like proteins in the mouse genome, *Opn4* is the most highly expressed in the iris. Interestingly, it is expressed in both dilator and constrictor muscles, and it is also expressed in iris PE cells (*Figure 2—figure supplement 1A*). Transcripts for cryptochromes-1 and -2 (*Cry1* and *Cry2*), flavin-based light sensors implicated in circadian rhythms, are also widely expressed in the iris (*Figure 2—figure supplement 1A*). Transcripts for PLC isozymes show multiple PLC isoforms in iris smooth muscle with *Plcb4* being the most abundant (*Figure 2—figure supplement 1B*). The abundances of transcripts for calcium release channels (IP3 receptors [*Itpr1*, *Itpr2*, and *Itpr3*] and ryanodine receptors [*Ryr1*, *Ryr2*, and *Ryr3*]) implies that their order of importance is IP3R1>> IP3 R2> IP3 R3 and RYR3>> RYR2> RYR1 (*Figure 2—figure supplement 1C*). The distribution of adrenergic and muscarinic receptors implies that (1) dilator muscles receive adrenergic input principally via ADRA1A, (2) sphincter muscles receive cholinergic input principally via CHRM3 – consistent with the defect in pupil constriction observed in *Chrm3* knockout mice (*Matsui et al., 2000*) – and, to a lesser extent, via CHRM2, (3) dilator muscles receive cholinergic input via CHRM2, and (4) the pharmacology of sphincter 2 cells is intermediate between sphincter 1 and dilator cells because sphincter 2 cells express levels of alpha-adrenergic receptors and muscarinic acetylcholine receptors that are intermediate between those of sphincter 1 and dilator cells (*Figure 1D* and *Figure 2—figure supplement 1D*). The distribution of alpha-adrenergic receptor transcripts also implies that stroma 2 cells are sensitive to adrenergic ligands via ADRA1B receptors (*Figure 2—figure supplement 1D*).

## Epithelial and stromal cells

The inner surface of the iris is occupied by the iris PE, which is contiguous with the adjacent CBE (*Figure 1B*). Transcription factor ZIC1 localizes exclusively to the nuclei of these two epithelia (*Figure 4*). OTX1 and OTX2, two highly homologous transcription factors, are expressed in sphincter and dilator muscles (in different proportions) and OTX1 is additionally expressed in iris PE, as seen by immunostaining with an antibody that recognizes both proteins, referred to hereafter as 'OTX'

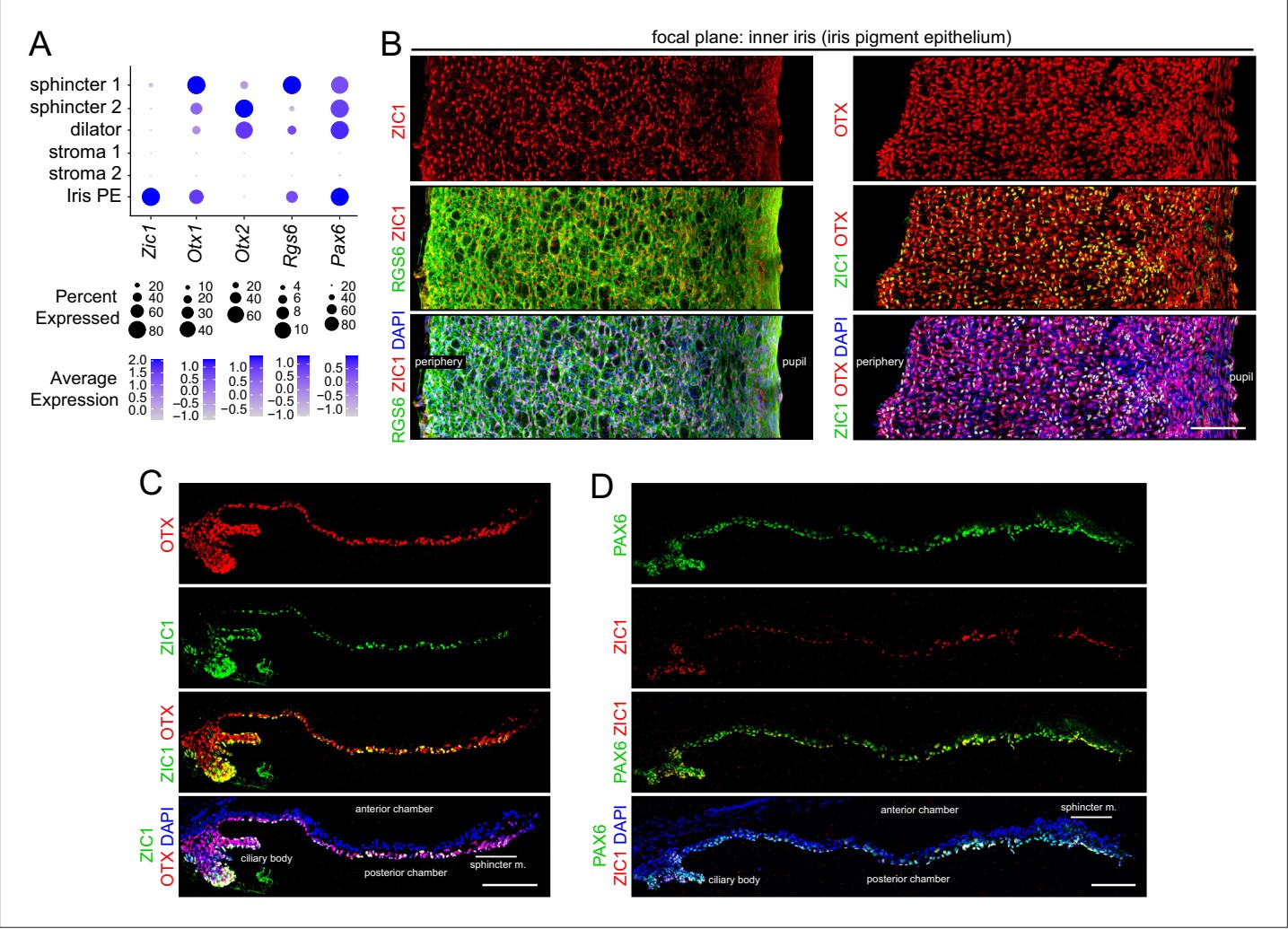

**Figure 4.** Immunostaining for iris epithelial markers. (**A**) Dot plot (as described in *Figure 2A*) showing transcript abundances across a subset of iris cell types for the markers shown in (**B–D**). (**B**) Flat mounts of untreated iris immunostained as indicated. (**C,D**) Cross sections of untreated iris immunostained as indicated. OTX refers to an antibody that binds to both OTX1 and OTX2. Scale bars, 100 μm.

(*Figure 4A–C*). The expression of PAX6, a master regulator of anterior eye development, closely matches the OTX expression pattern (*Figure 4A, C, and D*).

The snRNAseq data shows that stroma 1 cells express transcription factors *Sox10*, *Tbx1*, *Pax3*, and *Tfap2a*, whereas stroma 2 cells express *Pitx2*, *Glis1*, and *Tfap2b* (*Figures 1D, F and 5A*, and *Supplementary file 2*). For SOX10 and TFAP2B, these snRNAseq patterns were confirmed by immunostaining of iris flat mounts and sections, which shows their mutually exclusive localization in nuclei in the outer half of the iris, where stromal cells reside (*Figure 5B and C*). Co-staining of iris sections for SMA or PDGFRA, markers for muscle and stroma 2 cells, respectively, together with SOX10 or TFAP2B, and with DAPI, shows the trilayer iris structure: stromal cells in the outermost layer, dilator muscle in the central layer, and iris PE in the innermost layer (*Figure 5C*). Stroma 1 cells express multiple melanogenesis-related genes, including endothelin receptor beta (*Ednrb*), microphthalmia-associated transcription factor (*Mitf*), dopachrome tautomerase (dopachroma delta-isomerase, *Dct*), tyrosinase-related protein-1 (*Tyrp1*), and tyrosinase (*Tyr*), indicating that they serve as a secondary site of melanin synthesis in addition to the principal site of melanin synthesis and accumulation in the iris PE.

The distribution of immune cells in the rodent iris – principally macrophages and dendritic cells – has been defined previously by immunocytochemistry of iris flat mounts (*McMenamin, 1997*; *McMenamin, 1999*). We have extended these earlier observations by immunostaining for CD45 (leuko-cytes), CD206 (M2 macrophages), MHC class II antigens (I-A/I-E) (dendritic cells), PU.1 (macrophages,

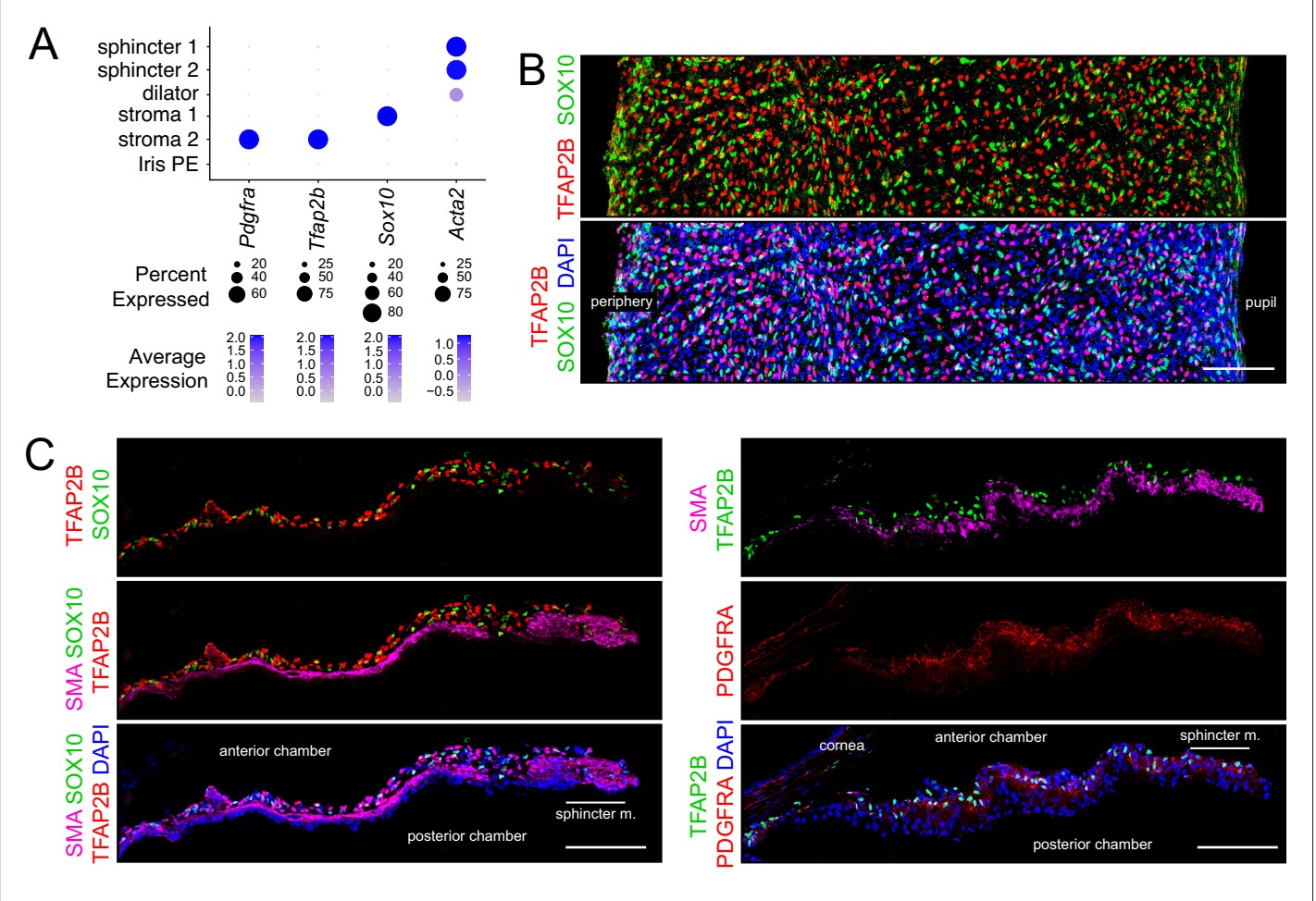

**Figure 5.** Immunostaining for stromal cell markers. (**A**) Dot plot (as described in *Figure 2A*) showing transcript abundances across a subset of iris cell types for the markers shown in (**B and C**). (**B**) Flat mounts of untreated iris immunostained as indicated. (**C**) Cross sections of untreated iris immunostained as indicated. All images are at the same magnification. Scale bars, 100 μm.

The online version of this article includes the following figure supplement(s) for figure 5:

**Figure supplement 1.** Immune cells in the iris.

neutrophils, and myeloid dendritic cells), and CLDN5 (blood vessels) (*Figure 5—figure supplement 1A*). The murine iris is uniformly tiled by irregularly shaped stellate cells, of which the majority are macrophages and the minority are dendritic cells (*Figure 5—figure supplement 1B*). The CD206 population (M2 macrophages) comprise approximately half of the CD45 population (*Figure 5—figure supplement 1C*).

## Effects of dilation and constriction on patterns of gene expression

One of the most striking properties of the iris is its mechanical flexibility. In the mouse iris, pharmacologically induced pupil dilation reduces the central-to-peripheral width of the iris by ~5 -fold, which produces a back-and-forth folding of nerves and blood vessels (*Figure 6A and B*). These large-scale changes in tissue structure suggested the possibility that iris dilation might also produce changes in gene expression.

As noted in connection with *Figure 1E* and *Figure 1—figure supplements 1 and 2*, the transcriptomes of constricted and untreated irises are virtually indistinguishable across all cell types, presumably because the untreated pupil diameter is relatively small, so that pharmacologic constriction produces only a modest additional reduction in its diameter. Therefore, comparisons between the transcriptomes of constricted and untreated irises can be used to estimate technical variability. Scatter

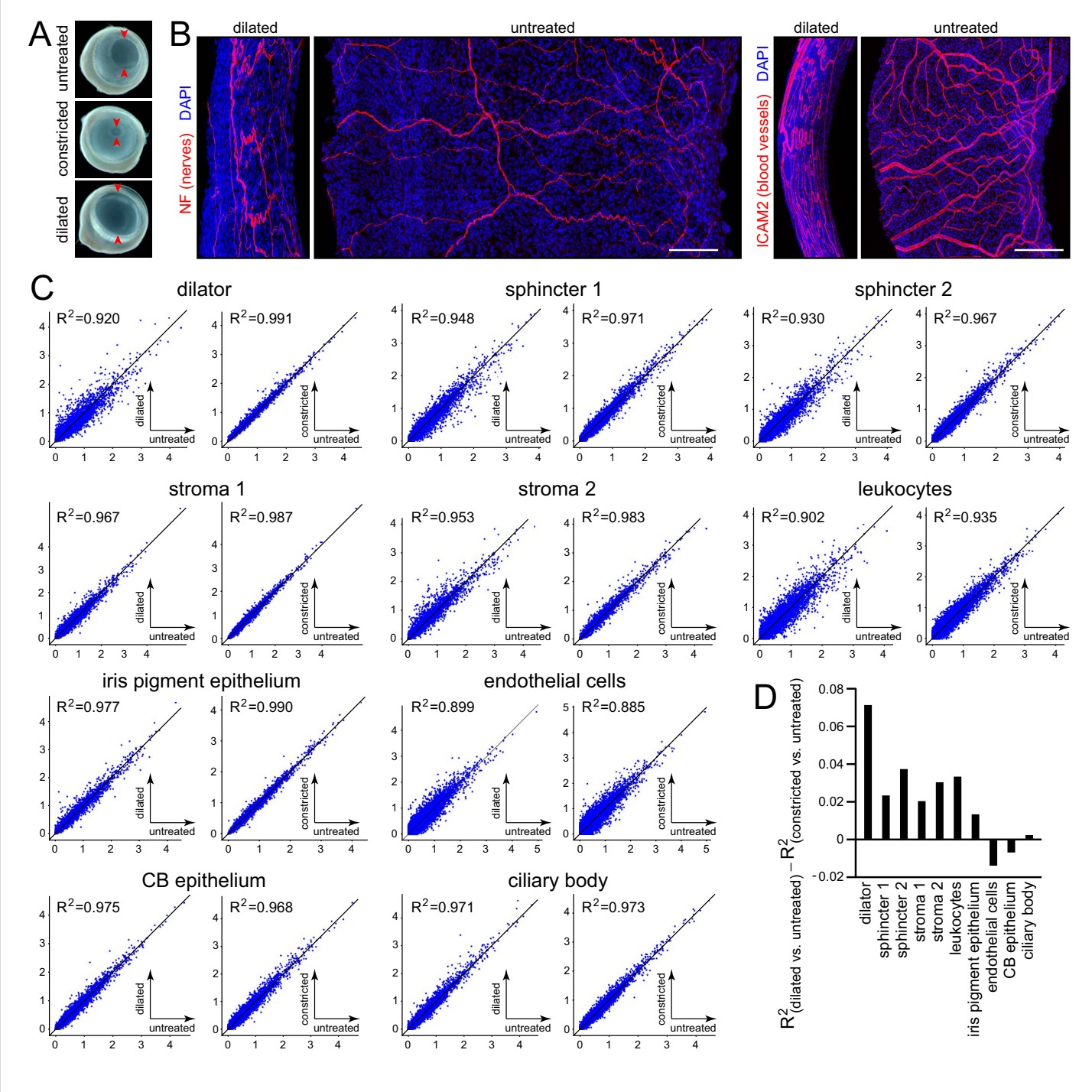

**Figure 6.** Morphological and transcriptomic changes between dilated and constricted/untreated iris. (**A**) Images of isolated eyes from mice with dilated pupils, constricted pupils, or no treatment. Red arrowheads mark the edge of the pupil. (**B**) Flat mounts of dilated and untreated irises showing the pattern of nerve fibers (left) and blood vessels (right). All images are at the same magnification and are oriented with the pupil to the right. Scale bars, 100 μm. (**C**) Scatter plots of single nucleus (sn)RNAseq read counts ($\log_{10}$ average normalized expression) and $R^2$ values for pairwise comparisons of all transcripts for dilated vs. untreated (left) and constricted vs. untreated (right) irises for each of the major iris and ciliary body cell types. (**D**) Summary plot showing the difference in $R^2$ values between the dilated vs. untreated scatter plot and the constricted vs. untreated scatter plot for each cell type.

The online version of this article includes the following figure supplement(s) for figure 6:

**Figure supplement 1.** Dot plot summary of transcriptome changes in dilated, constricted, and untreated irises.

**Figure supplement 2.** Volcano plots of single nuclues (sn)RNAseq read counts for dilated vs. untreated irises.

plots of snRNAseq read counts for the constricted vs. untreated comparisons showed deviations from $R^2 = 1$ that were correlated with sample size: the leukocyte and endothelial cell samples, which had the fewest cells (~50–200 cells per sample; *Supplementary file 1*), had the lowest $R^2$ values, ranging from 0.885 to 0.935, whereas dilator muscle, iris PE, and stroma 2 samples, which had the most cells (>1600 cells per sample; *Supplementary file 1*), had $R^2$ values between 0.983 and 0.991 (*Figure 6C*). For seven of the ten iris cell types, there was more scatter in the dilated vs. untreated comparison than in the constricted vs. untreated comparison (*Figure 6C and D*). The dilator muscle showed the largest transcriptome changes, with a reduction in $R^2$ from 0.991 (constricted vs. untreated) to 0.920 (dilated vs. untreated). The three cell types that showed little change in $R^2$ or a small trend in the opposite direction were endothelial cells, CB epithelial cells, and CB cells. Examples of transcripts that increased or decreased with dilation (defined as a $\log_2$-fold change greater than 0.6 or smaller than –0.6 and an adjusted p-value less than 0.05) were identified for nine of the ten cell types, as shown in *Figure 6—figure supplements 1 and 2* (*Supplementary file 3*; differential transcripts in endothelial cells did not reach these threshold criteria). The dot plots in *Figure 6—figure supplement 1* also illustrate the near identity of constricted and untreated snRNAseq read counts.

To independently assess the changes seen with snRNAseq, we focused on three dilator muscle transcripts that were among the most strongly induced by dilation: *Egr1*, a zinc-finger transcription factor that is an immediate-early response gene in diverse cell types; *Slc26a4*, a broad-specificity anion exchanger that is expressed in multiple epithelia; and *Tmem158*, a membrane protein of unknown function (*Figure 7A*). Immunostaining for EGR1, ZIC1, and SMA shows that EGR1 is undetectable in untreated iris and is readily detected in the nuclei of the dilator muscle with pupil dilation (*Figure 7B*). The same sections show that the abundances of ZIC1 in iris PE and SMA in muscle cells are unchanged by dilation. Similarly, ISH shows the accumulation of *Slc26a4* and *Tmem158* transcripts in dilator muscle with pupil dilation, whereas the abundance of *Adrala* transcripts remains unchanged (*Figure 7C*, quantified in *Figure 7—figure supplement 1*). Confirming the snRNAseq data, quantification of the ISH signals showed accumulation of *Anks1b*, *Btg2*, *Junb*, and *Pde10a* transcripts in dilator muscle and *Tmem158* transcripts in sphincter muscle with pupil dilation (*Figure 7—figure supplement 1*).

Among the transcripts that exhibit increased abundance with pupil dilation, *Pde10a*, which codes for a dual cAMP/cGMP phosphodiesterase (*Soderling et al., 1999*), stands out as showing the highest fold change in dilator, sphincter 2, stroma 1, iris PE, CB, CB epithelial cells, and it shows close to the highest fold change in the other iris cell types (*Figure 6—figure supplements 1 and 2*; *Supplementary file 3*). Other cyclic nucleotide phosphodiesterase transcripts – including *Pde1c*, *Pde3b*, *Pde4d*, *Pde7b* – are also upregulated, suggesting that negative regulation of cyclic nucleotide signaling is a general feature of iris dilation (*Supplementary file 3*). In dilator muscle, the second most highly induced transcript is *Btg2*, and, unlike *Pde10a*, *Btg2* induction is specific to dilator muscle (*Supplementary file 3*). BTG2 interacts with the CAF1 subunit of the CCR4-NOT complex to promote deadenylation of mRNAs, with effects ranging from countering cardiomyocyte hypertrophy to maintaining T cell quiescence (*Mauxion et al., 2008*; *Yang et al., 2008*; *Masumura et al., 2016*; *Hwang et al., 2020*). BTG2 could play a regulatory role in the iris dilator muscle analogous to its role in cardiac muscle.

The rapid accumulation of *Egr1* transcripts and EGR1 protein (also called ZIF268/ZNF268 and KROX24) in contexts as diverse as serum-stimulated cell proliferation (*Christy et al., 1988*) and neural activity (*Worley et al., 1991*) suggested that a similarly rapid response might be observed in the dilated iris. A time course experiment in which irises were harvested 30, 90, or 150 min after the onset of pupil dilation (maintained by hourly instillations of phenylephrine + cyclopentolate eye drops) showed that EGR1 accumulation in the dilator muscle was apparent within 30 min of the onset of dilation and was maintained thereafter with continuous pupil dilation (*Figure 7D*). At the 30 min point, there were also scattered EGR-1[+] cells in the stromal layer, but the numbers of these cells declined at later time points (compare *Figure 7A and D*).

## Effects of dilation on nuclear morphology

Changes in iris structure at the macroscopic level must have counterparts at the cellular level. Early electron micrographic studies visualized ultrastructural responses to dilation or constriction on the plasma membrane morphology of iris PE cells and the surface morphology of sphincter and dilator

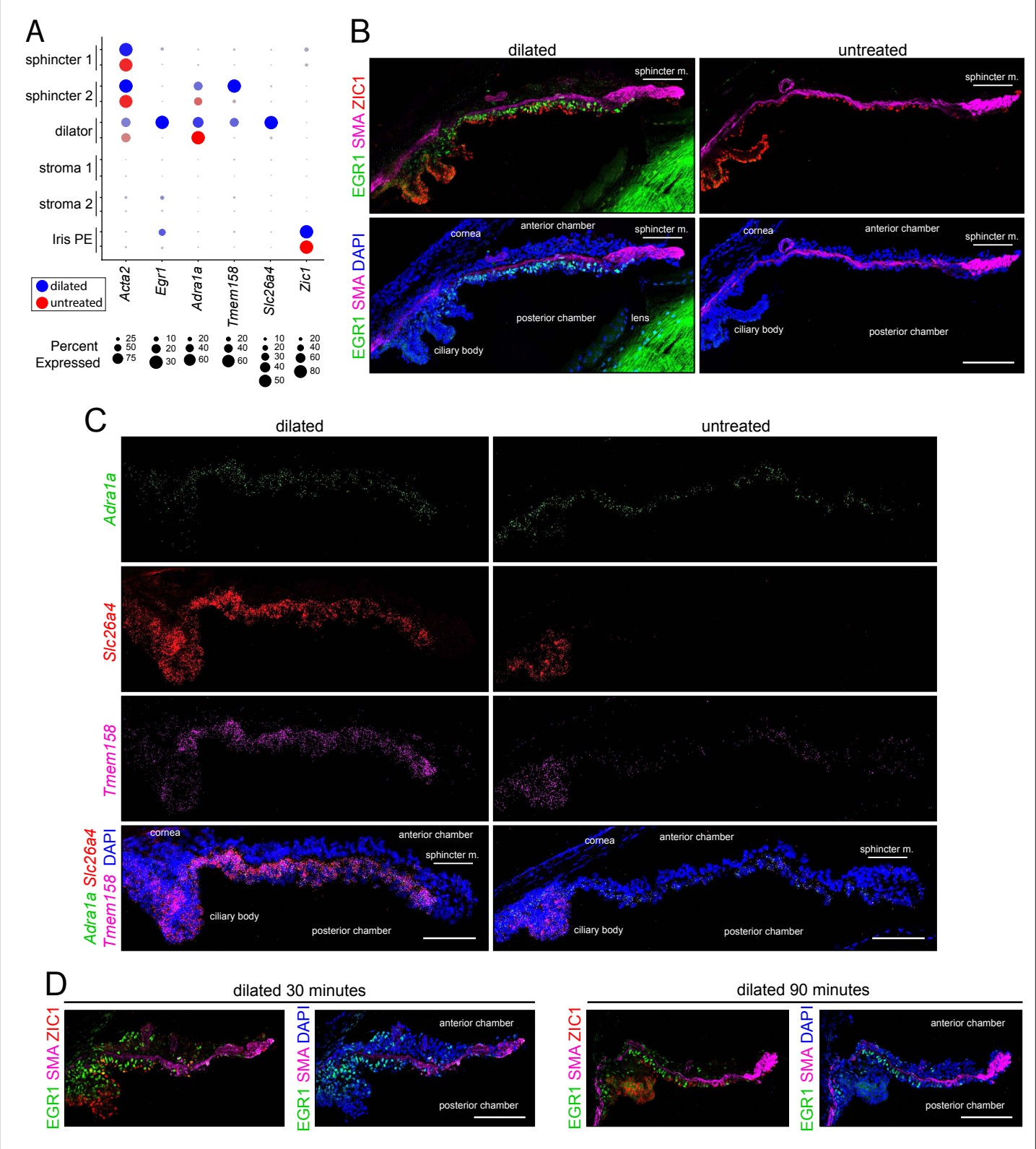

**Figure 7.** Immunostaining and in situ hybridization for markers that distinguish dilated vs.untreated irises. (**A**) Dot plot (as described in *Figure 2A*) showing transcript abundances across a subset of iris cell types for the markers shown in (**B and C**). For each cell type, the symbols for the dilated iris (blue) are above the symbols for the untreated iris (red). (**B**) Cross sections of dilated (left) and untreated (right) irises immunostained as indicated. (**C**) Cross sections of dilated (left) and untreated (right) irises hybridized with the indicated probes. (**D**) Immunostaining of irises harvested 30 min (left) and

*Figure 7 continued on next page*

Figure 7 continued

90 min (right) after the onset of dilation. All images are at the same magnification. Scale bars, 100 μm.

The online version of this article includes the following figure supplement(s) for figure 7:

**Figure supplement 1.** Quantification of in situ hybridization (ISH) signals in untreated vs. dilated irises.

muscles (*Lim and Webber, 1975a*; *Lim and Webber, 1975b*; *Murata et al., 1998*). Still largely unexplored are the effects of changes in iris structure on nuclear morphology. Current evidence suggests that changes in nuclear morphology alter chromatin organization, gene expression, and intracellular signals (*Kirby and Lammerding, 2018*; *Lele et al., 2018*).

The identification of (1) transcription factors that mark defined iris cell type and (2) antibodies that can be used to immunolocalize those proteins in iris whole mounts presented an opportunity to visualize nuclear morphology in defined cell types in response to changes in iris structure (*Figure 8A*). Nuclei were visualized from the following cell types (with the corresponding marker in parenthesis): endothelial (ERG), sphincter 1 (NeuN), stroma 1 (SOX10), stroma 2 (TFAP2B), iris PE (ZIC1), and dilator muscle (OTX but not ZIC1). For untreated, dilated, and constricted irises, the length:width ratio was quantified for fluorescently stained nuclei in Z-stacked confocal images of iris flat mounts (*Figure 8B* and ). 'Length' refers to the longest dimension and 'width' refers to the dimension perpendicular to length, both within the X-Y plane of the flat mounted iris. As the 2D projection of a 3D ellipsoid exhibits a length:width ratio less than or equal to the actual 3D ratio, our analysis may underrepresent the elongation of some nuclei.

The nuclear length:width ratio comparison shows no significant changes with dilation in the highly elongated nuclei of sphincter 1 cells (mean length:width ratio ~6). However, all other cell types showed highly significant ($p < 0.0001$) increases in nuclear elongation with dilation. The smallest changes were seen with OTX$^+$ nuclei, in which the mean length:width ratio increased from ~2 to ~3. Stroma cell nuclei (SOX10$^+$ and TFAP2B$^+$) showed the largest increases in the mean length:width ratio, from ~2 to ~5. The biological effects of these changes in nuclear morphology remain to be determined.

## Iris lineage tracing

Lineage tracing experiments in avian embryos provided the first evidence that some anterior ocular structures were derived, at least partially, from the neural crest via its contribution to the peri-ocular mesenchyme (*Johnston et al., 1979*). Neural crest lineage tracing in mice using *Cre-LoxP* to mark cells has produced conflicting results, with one laboratory describing no contribution of neural crest cell to the iris with a *Wnt1-Cre* driver (*Gage et al., 2005*) and a second laboratory showing neural crest contributions to iris stroma with a *P0-Cre* driver (*Kanakubo et al., 2006*; *Kikuchi et al., 2011*).

We have revisited this issue using *Sox10-Cre* (*Matsuoka et al., 2005*) and *Wnt1-Cre2* (*Lewis et al., 2013*) drivers, a nuclear-localized histone *H2B-GFP* reporter (*R26-LSL-tdT-2A-H2BGFP*; *Wang et al., 2018*), and immunostaining for the iris cell-type-specific transcription factors characterized in this study (*Figure 9* and *Figure 9—figure supplement 1*). *Sox10-Cre* labels migratory neural crest cells and *Wnt1-Cre2* labels multiple cells within the embryonic CNS, including premigratory neural crest (*Debbache et al., 2018*; *Keuls and Parchem, 2021*). For each *Cre* driver line, the fraction of cells of a given type that expressed the *H2B-GFP* reporter was determined from iris flat mounts and is summarized in *Figure 9A*. As seen in the iris flat mount and cross-sectional images in *Figure 9* and *Figure 9—figure supplement 1*, co-labeling with H2B-GFP was widespread with the *Wnt1-Cre2* driver but limited to stroma 1 and stroma 2 cells with the *Sox10-Cre* driver. [Independent of lineage, the labeling of stroma 1 cells with the *Sox10-Cre* driver is expected, since these cells express *Sox10* in the mature iris.] The only cell type that showed less than a 50 % contribution from *Wnt1-Cre2*-labeled cells was sphincter 1 (*Figure 9C*). The data are consistent with a model in which (1) a subset of migratory neural crest cells, defined by *Sox10-Cre* expression, contribute exclusively to the stromal cell populations and provide all or nearly all of the stromal cell progenitors, and (2) migratory neural crest cells that express *Wnt1-Cre2* but not *Sox10-Cre* contribute the majority of the progenitors for iris PE, dilator, and sphincter 2 cells, but contribution minimally to the sphincter 1 population. The vast majority of sphincter 1 cells presumably arise either from an unlabeled neural crest population or from local ocular progenitors.

## Discussion

The work described here is presented as a resource for future investigations of the mammalian iris. We have (1) used snRNAseq to define all of the major cell types in the mouse iris, which has led to the discovery of two types of stromal cells and two types of sphincter cells; (2) validated a series of molecular markers that can be used to visualize each of the major iris cell types; (3) identified transcriptome changes and distortions in nuclear morphology associated with iris dilation; and (4) clarified the neural crest contribution to the iris by showing that *Wnt1-Cre*-expressing progenitors contribute to nearly all iris cell types, whereas *Sox10-Cre*-expressing progenitors contribute only to stromal cells. As described more fully below, this work should be useful as a point of reference for investigations

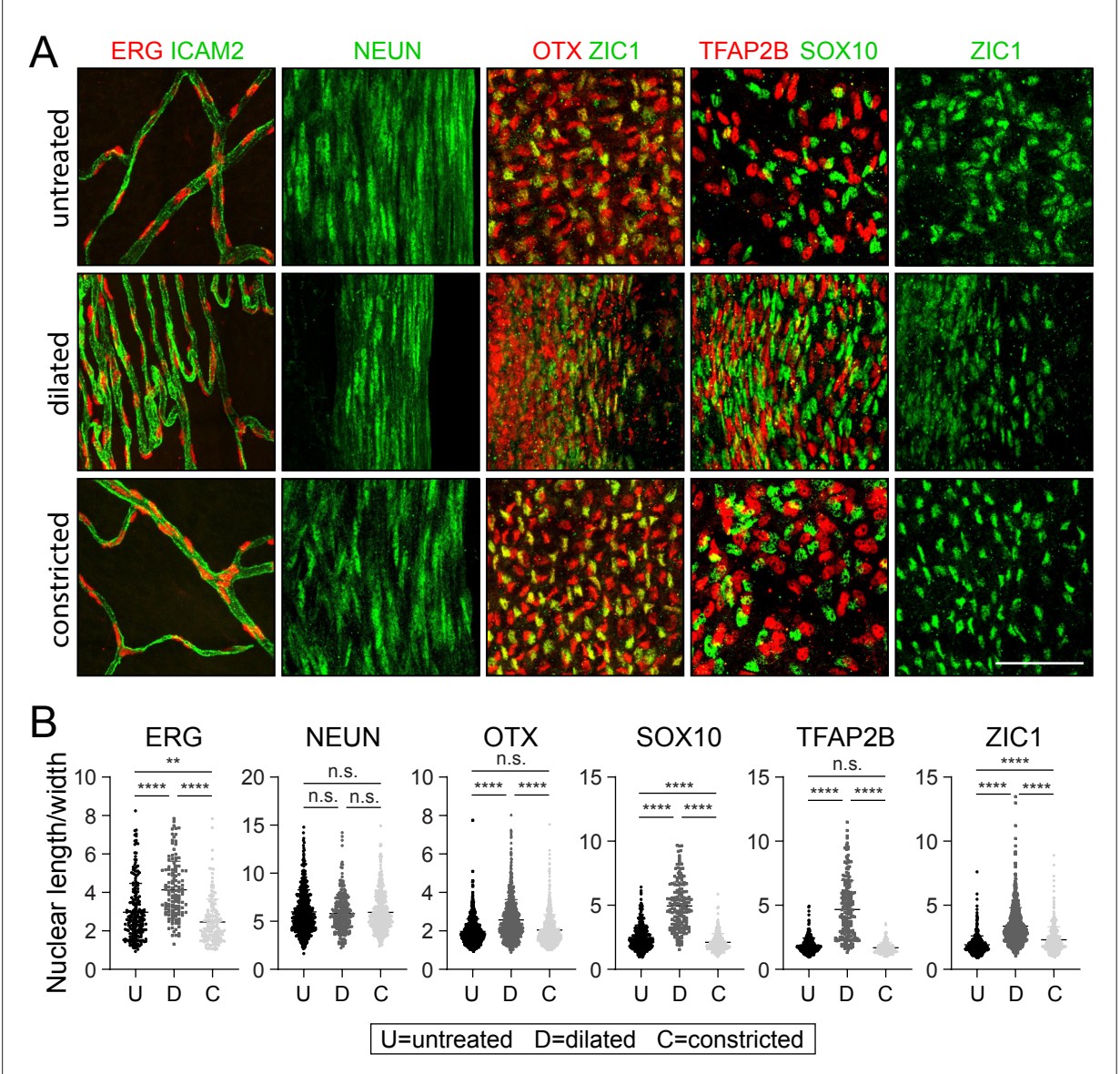

**Figure 8.** Changes in nuclear morphology in response to dilation or constriction for defined iris cell types. (**A**) Flat mounts of untreated, dilated, or constricted irises, immunostained for the indicated nuclear markers. All images are at the same magnification. Scale bar, 50 μm. (**B**) Quantification of the length/width ratio for individual nuclei viewed in iris flat mounts. Cell types are defined by the indicated marker. For the OTX category, only OTX$^+$ and ZIC1$^-$ cells from the OTX plus ZIC double-labeled images were scored. Each data point is an individual nucleus. The data for each condition in each plot are derived from three mice. **p < 0.01; ****p < 0.0001; n.s., not significant. Bars, mean ± SD.

The online version of this article includes the following figure supplement(s) for figure 8:

**Figure supplement 1.** Quantification of the length/width ratio for individual nuclei viewed in iris flat mounts.

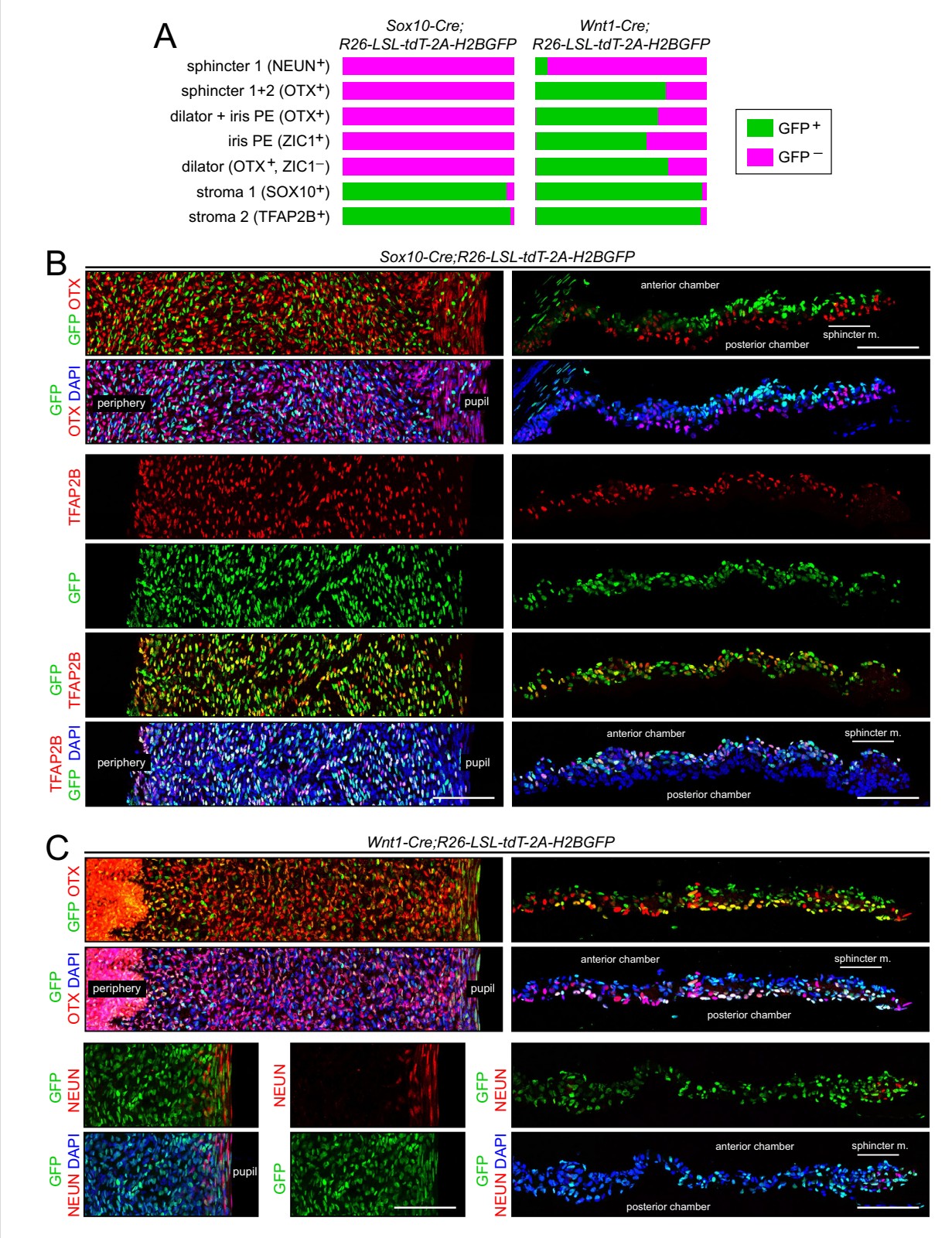

**Figure 9.** Lineage tracing of adult iris cell types with *Sox10-Cre* and *Wnt1-Cre*. (**A**) Bar graphs showing the fraction of cells that express nuclear-localized H2B-GFP, indicating excision of the *LoxP-stop-LoxP* cassette from the *Cre* reporter *R26-LSL-tdT-2A-H2BGFP*. Iris cell types were identified by immunostaining for the nuclear-localized markers listed. The intensities of GFP and the cell type markers vary among nuclei. A nucleus was considered to exhibit co-localization if both signals were present, regardless of intensity. For sphincter muscle, the number of cells scored for each *Cre* driver ranged

*Figure 9 continued on next page*

Figure 9 continued

from 120 to 140. For each of the other cell types, the number of cells scored for each *Cre* driver ranged from 960 to 2300. (**B**) Iris flat mounts (left) and cross sections (right) from *Sox10-Cre;R26-LSL-tdT-2A-H2BGFP* mice showing mutually exclusive localization of GFP and OTX1 +2 ('OTX') (upper panels) and partial co-localization of GFP and TFAP2B (lower panels). (**C**) Iris flat mounts (left) and cross sections (right) from *Wnt1-Cre;R26-LSL-tdT-2A-H2BGFP* mice showing partial co-localization of GFP and TFAP2B (upper panels) and minimal co-localization of GFP and NeuN (lower panels). All images are at the same magnification. Scale bars, 100 µm.

The online version of this article includes the following figure supplement(s) for figure 9:

**Figure supplement 1.** Lineage tracing of stroma 1 cells with *Sox10-Cre;R26-LSL-tdT-2A-H2BGFP*.

of iris development, disease, and pharmacology, for the isolation and propagation of defined iris cell types, and for iris cell engineering and transplantation. Going forward, it will be of interest to obtain and compare similar snRNAseq data from the irises of other species, most especially from humans.

## Iris cell types

The present work defines the molecular heterogeneity among different classes of iris smooth muscle cells. Smooth muscle cells are present in multiple organs and they control a wide range of physiological functions, including vascular tone, airway resistance, gastrointestinal (GI) motility, and pupil diameter. In contrast to the unitary mode of skeletal muscle excitation – quantal release of acetylcholine at the neuromuscular junction leading to activation of postsynaptic nicotinic receptors and depolarization of the muscle membrane – smooth muscle activation and inhibition involve multiple classes of GPCRs that activate or inhibit calcium mobilization and protein phosphorylation/dephosphorylation (*Kuo and Ehrlich, 2015*). The comprehensive determination of the abundances of transcripts for all known receptors, channels, and signaling components for sphincter 1, sphincter 2, and dilator muscles constrains the possible ligand-receptor systems that control contraction and relaxation in each of these muscle types.

The iris PE cell transcriptome provides a foundation for strategies to genetically engineer these cells and monitor their state of trans-differentiation in cell culture. Over the past 30 years, the iris PE has been studied as a potential source of cells for autologous transplantation to replace dying or dysfunctional retinal pigment epithelial (RPE) cells in individuals with age-related macular degeneration. These studies were motivated by the surgical accessibility of human iris PE cells, which can be harvested from a small iridectomy sample and expanded in culture, and by earlier work showing that the iris PE in non-mammalian vertebrates can transdifferentiate into other ocular cell types (*Hu et al., 1997*; *Sun et al., 2006*). In three clinical trials of sub-retinal transplantation of iris PE in age-related macular degeneration patients, there have been minimal complications but also minimal effect on the clinical course of the disease (*Lappas et al., 2004*; *Aisenbrey et al., 2006*; *Abe et al., 2007*). The failure to significantly alter disease progression is likely referable to the failure of the transplanted iris PE cells to acquire the phagocytic and other properties of RPE cells, a deficiency that might be remedied if iris PE cells can be genetically engineered to trans-differentiate toward an RPE fate.

The identification of two types of iris stromal cells adds to a growing literature on molecular diversity among fibroblasts and stromal cells that has emerged from single-cell RNA sequencing (*Guerrero-Juarez et al., 2019*; *DeSisto et al., 2020*; *Biffi and Tuveson, 2021*). One extension of the present study will be to define the responses of stroma 1 and stroma 2 cells to inflammation. Iris inflammation (iritis) is one component of anterior segment uveitis, a common autoimmune condition (*Martin et al., 2002*; *Artornsombudh et al., 2014*; *Wakefield et al., 2020*). Work in a variety of disease contexts has shown that stromal cells change their patterns of gene expression in response to inflammation and that they can epigenetically 'remember' prior bouts of inflammation (*Neumann et al., 2010*; *Crowley et al., 2018*; *Mizoguchi et al., 2018*). The chronic and relapsing/remitting character of uveitis suggests that iris stromal cells might similarly undergo long-lasting changes in gene expression during a bout of uveitis (*Grunwald et al., 2011*).

## Relevance to ocular disease

The ease with which the iris can be visualized by the examining physician and the large effect that anatomic or functional defects in the iris have on retinal image quality facilitates the identification of individuals with inherited or acquired defects in iris structure. In the paragraphs that follow, we summarize a set of iris disorders that arise from defects in smooth muscle, iris PE, iris stroma, or

migrating neural crest cells. For each of these examples, the cell-type-specific transcriptome changes that occur in the affected iris have not yet been determined.

The most severe inherited disorders of smooth muscle cell function, referred to as multisystemic smooth muscle dysfunction syndrome (MSMDS) or megacystis-microcolon-intestinal hypoperistalsis syndrome (MMIHS), manifest in infancy with life-threatening defects in GI motility and are associated with mutations in the genes coding for components of the contractile machinery, including myosin light chain kinase (*MYLK*), SMA (*ACTA2*), and myosin heavy chain-11 (*MYH11*) (*Moreno et al., 2016*; *Gamboa and Sood, 2019*; *Hashmi et al., 2021*). Defects in pupil constriction have been reported in patients with MSMDS/MMIHS, consistent with contractile defects in iris smooth muscle (*Moller et al., 2012*; *Roulez et al., 2014*). Going forward, it will be of interest to assess pupil function and to look for genetic variation in smooth muscle genes in individuals with less severe defects in GI motility, bladder emptying, uterine contraction, and other smooth muscle functions.

The release into the aqueous humor of excessive quantities of cellular material, including pigment granules, from the iris PE is the hallmark of pigment dispersion syndrome (PDS) (*Niyadurupola and Broadway, 2008*). If sufficient quantities of this material become trapped in the trabecular meshwork, the ocular drainage structure, the resulting rise in intraocular pressure can lead to a subtype of open angle glaucoma referred to as pigmentary glaucoma (PG). PDS is associated with increased contact between the iris and lens, which may predispose to pigment release secondary to mechanical trauma (*Aptel et al., 2011*). In mice, mutations in melanosomal genes lead to PDS, PG, and a related disorder, iris stromal atrophy (*Anderson et al., 2002*). In humans, PDS also has a genetic component, but the contributing genes are still largely unknown (*Tandon et al., 2019*; *van der Heide et al., 2021*). A conceptually similar disorder, exfoliation syndrome, is characterized by impaired pupil constriction and by the release of extracellular matrix material from the iris and lens, which accumulates in the trabecular meshwork and can lead to open angle glaucoma (*Kivelä, 2018*; *Schlötzer-Schrehardt, 2018*; *Tekin et al., 2020*). Exfoliation syndrome has a strong genetic component, with seven loci identified in genome-wide association studies (*Aboobakar et al., 2017*).

The iris stroma and PE are sites of cyst formation, which can be either congenital or acquired. Most primary cysts are benign and do not lead to visual disability. Cysts that arise as secondary consequences of surgical or non-surgical trauma are more likely to enlarge and can lead to obstruction of vision, uveitis, or glaucoma (*Shields and Shields, 2017*; *Georgalas et al., 2018*). It is not currently known whether the cells comprising iris cysts carry mutations that promote aberrant growth, as has been observed in cysts in several other organs (*Wu et al., 2011*; *Wang et al., 2016*; *Napolitano et al., 2020*).

Consistent with the migratory origin of multiple iris cell types, the iris is sensitive to defects in neural crest migration (*Weigele and Bohnsack, 2020*). Transcription factors PITX2 and FOXC1 are required for normal migration of peri-ocular mesenchyme cells, and mutations in *PITX2* or *FOXC1* produce (1) hypoplasia of the iris stroma, holes in the iris, and an eccentric position of the pupil (Rieger anomaly), and/or (2) strands of ectopic tissue in the space between iris and cornea (Axenfeld anomaly). Iris development is also sensitive to reductions in *PAX6* activity, with phenotypes ranging from mild iris hypoplasia to complete aniridia (*Hingorani et al., 2012*; *Cvekl and Callaerts, 2017*).

An important iatrogenic disorder, intraoperative floppy iris syndrome (IFIS), occurs during ~2 % of cataract surgeries and is a risk factor for surgical complications (*Enright et al., 2017*; *Christou et al., 2020*). IFIS is characterized by pupil constriction, a flacid and billowing iris, and prolapse of the iris through the surgical incision. IFIS is strongly linked to prior or ongoing systemic treatment with alpha1-adrenergic receptor antagonists, most commonly for benign prostatic hypertrophy, which presumably leads to long-term changes in iris smooth muscle physiology.

## Gene expression and morphologic responses to dilation

The asymmetry in gene expression responses, in which an iris with a dilated pupil differs from an untreated iris but an iris with a constricted pupil is nearly identical to an untreated iris, most likely reflects the tissue compression associated with pupil dilation, as seen in the zig-zagging morphology of blood vessels and nerves (*Figure 6B*), the flattening of nuclear shape (*Figure 8*), and the membrane infoldings demonstrated in ultrastructural studies (*Lim and Webber, 1975a*; *Lim and Webber, 1975b*). By contrast, with pupil constriction, the iris is flattened to an even greater extent than it is in the untreated (resting) state.

The gene expression changes observed with pupil dilation could represent physiologic adaptations to changes in metabolic state and/or cell shape. Consistent with this view, *Pde10a*, a transcript that is highly expressed in the brain and is strongly induced by dilation in iris and ciliary body cell types, codes for a cyclic nucleotide phosphodiesterase that may function to homeostatically regulate cAMP and/or cGMP signaling (*Conti and Beavo, 2007*; *Wilson and Brandon, 2015*). Similarly, *Egr1*, one of the transcripts that is strongly induced in dilator muscle, codes for an immediate early transcription factor that responds to a wide variety of physiologic stimuli, including growth factor stimulation in many cell types and synaptic activity in neurons (*Duclot and Kabbaj, 2017*). Extrapolating from the observations reported here to the human iris, it seems plausible that iris gene expression changes might also accompany the pharmacologic pupil dilation associated with routine ophthalmoscopic examinations of the human retina.

Nuclear elongation in response to dilation is most apparent in non-muscle cells, where nuclei in the resting and constricted states are minimally elongated. The nucleus is the largest cellular organelle, and distortions in nuclear shape in response to mechanical stress have been studied in systems ranging from diapedesis of immune cells to contraction of cardiac and skeletal muscle (*Kirby and Lammerding, 2018*). Changes in the force exerted on the nucleus lead to (1) changes in transcription factor localization, as seen for YAP/TAZ (*Elosegui-Artola et al., 2017*; *Kassianidou et al., 2019*), (2) chromatin reorganization (*Miroshnikova et al., 2017*), (3) direct and indirect changes in gene expression (*Tajik et al., 2016*), and (4) changes in cytoskeletal organization (*Hoffman et al., 2020*).

The optical accessibility of the iris offers a unique opportunity to extend the study of nuclear mechanobiology by imaging of nuclear dynamics and signal transduction in an intact tissue. For example, with the appropriate promoters and transgenes, stroma 1 or stroma 2 nuclei could be selectively visualized with a nuclear-localized far-red fluorescent protein to observe how their shapes change as the iris dilates and constricts in vivo or ex vivo. Reporters for other cellular structures (e.g., cytoskeleton) or molecules (e.g., calcium) could be similarly imaged in a cell-type-specific manner.

## Materials and methods
### Mice
The following mouse lines were used: *Wnt1-Cre2* (JAX 022501; *Lewis et al., 2013*), *Sox10-Cre* (JAX 025807; *Matsuoka et al., 2005*), and *R26-LSL-tdTomato-2A-H2B-GFP* (*Wang et al., 2018*). All mice were housed and handled according to the approved Institutional Animal Care and Use Committee protocol of the Johns Hopkins Medical Institutions. Iris snRNA-seq experiments and histological studies were carried out with 6- to 8 -week-old female albino mice. For lineage tracing experiments *Sox10-Cre*, *Wnt1-Cre2*, and *R26-LSL-tdTomato-2A-H2BGFP* transgenes were first crossed into an albino background. Genotyping primers were as follows: *Wnt1-Cre2* and *Sox10-Cre* (generic *Cre* primers), *Cre*-F, 5′-TGCCACGACCAAGTGACAGCAATGCTGTTT-3′ and *Cre*-R, 5′-ACCAGAGACGGA AATCCATCGCTCGACCAG-3′. *R26-LSL-tdTomato-2A-H2B-GFP*, *CAG*-F, 5′-CTAGAGCCTCTGCTAA CCATG-3′; *LSL*-R, 5′-CCTCTACAAATGTGGTATGGCTG-3′.

### Antibodies and other reagents
The following antibodies were used: mouse mAb anti-SMA, FITC conjugated (Sigma F3777); mouse mAb anti-SMA, Cy3-conjugated (Sigma C6198); rabbit anti-TFAP2B (Novus Biologicals NBP1-89063); goat anti-SOX10 (Santa Cruz Biotechnology SC-17342); rabbit anti-SOX10 (Cell Signaling Technologies 89356); chicken anti-GFP (Abcam Ab13970); goat anti-ZIC1 (R&D system AF4978-SP); rabbit anti-NeuN (Cell Signaling Technologies 24307 S); goat anti-CD45 (R&D systems AF114-SP); rat anti-PU.1 (Novus Biologicals MAB7124-SP); rat anti-CD206 (Bio-Rad MCA2235T); rat anti-MHCII (I-A/I-E) (Thermo Fisher Scientific/eBioscience 14-5321-82); rat anti-ICAM2 (BD Biosciences 553326); rabbit anti-Neurofilament 200 (Sigma N4142); chicken anti-SYT1 (Abcam ab133856); rabbit anti-NPTX2 (a gift from Dr Paul F Worley); rabbit anti-NOS1 (Cell Signaling Technologies 4231); rabbit anti-OTX2 (Proteintech 13497–1-AP); rabbit anti-PAX6 (a gift from Dr Randall Reed); rabbit anti-TTN (Proteintech 27867–1-AP); rabbit anti-CKM (MyBioSource MBS2006460); rabbit anti-CTNNA3 (Proteintech 13974–1-AP); rabbit anti-EGR1 (Cell Signaling Technologies 4154); rabbit anti-ERG (Cell Signaling Technologies 97249); rabbit anti-RGS6 (a gift from Dr Rory A Fisher); mouse Alex-Fluor 488-conjugated mAb 4C3C2 anti-CLDN5 (Thermo Fisher Scientific 352588). Alexa-Fluor-conjugated secondary antibodies

were from Invitrogen. Eye drops for dilation or constriction were prepared with (R)-(-)-phenylephrine (Sigma P6126); cyclopentolate (Sigma C-5165); and pilocarpine (Sigma P6503).

## Pupil dilation and constriction

For continuous pupil dilation, several microliters of 5 % phenylephrine +1 % cyclopentolate in 0.9 % NaCl (adjust with NaOH to pH = 7.5) was added to each eye once per hour for 6 hr; the mice were kept in the dark during the procedure. For continuous pupil constriction, several microliters of 5 % pilocarpine was added to each eye once per hour for 6 hr; the mice were kept in the light during the procedure. Thirty minutes after the last dose, the mice were deeply anesthetized and the eyes were enucleated for further processing.

## Tissue processing and immunostaining

Mice were deeply anesthetized with ketamine and xylazine, and then euthanized by cervical dislocation. For iris flat mounts, intact eyes were immediately enucleated and immersed in 2 % paraformaldehyde (PFA) in PBS at room temperature for 1 hr. Following immersion fixation, each eye was washed three times in PBS, the posterior of the eye was incised, the lens was carefully removed, and the iris was excised by cutting circumferentially around its margin. The intact irises were incubated overnight with primary antibodies diluted in PBSTC (PBS + 0.5 % Triton X-100, 0.1 mM $CaCl_2$) plus 10 % normal goat serum. Incubation and washing steps were performed at 4 °C. Tissues were washed four times with PBSTC over the course of 6–8 hr, and subsequently incubated overnight with secondary antibodies diluted in PBSTC +10 % normal goat serum. If a primary rat antibody was used, secondary antibodies were additionally incubated with 0.5 % normal mouse serum as a blocking agent. The next day, tissues were washed at least four times with PBSTC over the course of 6 hr, flat-mounted on Superfrost Plus glass slides (Fisher Scientific), and coverslipped with Fluoromount G (EM Sciences 17984-25).

For iris cross sections, eyes were embedded in optimal cutting temperature compound (OCT, Tissue-Tek), rapidly frozen in dry ice, and stored at –80 °C; 18 μm cross sections were cut on a cryostat and thaw-mounted onto Superfrost plus slides. Slides were stored at –80 °C until processing. Sections were immersed in 2 % PFA in PBS at room temperature for 15 min, washed three times in PBS and incubated overnight with primary antibodies diluted in PBSTC plus 10 % normal goat serum at 4 °C. The following day, sections were washed at least four times with PBSTC and incubated with secondary antibodies for 2 hr at room temperature. Sections were then washed four times with PBSTC and coverslipped with Fluoromount G. For each immunostaining analysis, iris cross sections were stained from at least five mice and iris whole mounts were stained from at least two mice.

## snRNAseq

Five or six irises were used for each snRNAseq library and two or three independent biological replicate libraries were prepared for each condition (*Supplementary file 1*). The eight groups of irises were collected and processed with the 10 × Genomics protocol in five batches, and the libraries were sequenced in two runs. The different conditions (constricted, dilated, and untreated) were each distributed across two or more batches and libraries from each condition were present in each of the two sequencing runs. Subsequent snRNAseq nucleus counts show that the percent of cells derived from CB and CBE to be tightly correlated within each sample ($R^2 = 0.9$) and together they vary from 3% to 15% of the total cell number – presumably reflecting variable inclusion of the ciliary body as a result of variability in cutting around the margin of the iris. Nucleus counts for other cell types, which are intrinsic to the iris, showed less variation between samples.

Irises were rapidly dissected in ice-cold DPBS (Gibco 14287072). The tissue was minced with a razor blade and Dounce homogenized using a loose-fitting pestle in 5 ml homogenization buffer (0.25 M sucrose, 25 mM KCl, 5 mM $MgCl_2$, 20 mM Tricine-KOH, pH 7.8) supplemented with 1 mM DTT, 0.15 mM spermine, 0.5 mM spermidine, EDTA-free protease inhibitor (Roche 11836 170 001), and 60 U/mL RNasin Plus RNase Inhibitor (Promega N2611). A 5 % IGEPAL-630 solution was added to bring the homogenate to 0.3 % IGEPAL CA-630, and the sample was further homogenized with five strokes of a tight-fitting pestle. The sample was filtered through a 50 μm filter (CellTrix, Sysmex, 04-004-2327), underlayed with solutions of 30% and 40% iodixanol (Sigma D1556) in homogenization buffer, and centrifuged at 10,000 × *g* for 18 min in a swinging bucket centrifuge at 4 °C. Nuclei were

collected at the 30–40% iodixanol interface, diluted with two volumes of homogenization buffer and concentrated by centrifugation for 10 min at 500 × *g* at 4 °C. snRNAseq libraries were constructed using the 10 × Genomics Chromium single-cell 3′ v3 kit and following the manufacturer's protocol (https://support.10xgenomics.com/single-cell-gene-expression/library-prep/doc/user-guide-chromium-single-cell-3-reagent-kits-user-guide-v31-chemistry). Libraries were sequenced on an Illumina NovaSeq 6000. snRNAseq data has been deposited in the GEO database (NIH), accession numbers GSE183690 and GSM5567780-GSM5567787.

## Analysis of snRNAseq data

Reads were aligned to the mm10 pre-mRNA index using the Cell Ranger count program version 3.1.0. The data for the different libraries was merged using Cell Ranger merge command. Data analysis was preformed using the Seurat R package version 4.0.1. After filtering out nuclei with >1% mitochondrial transcripts or with <500 or >6000 transcripts, 34,357 nuclei were retained. After the data was normalized using a regularized negative binomial regression algorithm (implemented in the SCTransform function as described in *Hafemeister and Satija, 2019*), there appeared to be little or no batch effects as judged by the high pairwise Pearson correlations (0.98–0.99 within each condition) and the highly similar UMAPs for each sample (see the new *Figure 1—figure supplements 1 and 2*). UMAP dimensional reduction was performed using the R uwot package (https://github.com/jlmelville/uwot; *Melville et al., 2020*) integrated into the Seurat R package. To compare cell types across treatments, the data was integrated using the strategy described in *Stuart et al., 2019*. This pipeline involves splitting the dataset by treatment using the Seurat SplitObject function and integrating the subset objects using FindIntegrationAnchors and IntegrateData functions. Data for the various scatter plots was extracted using the Seurat's AverageExpression function, and differential gene expression was analyzed using the Seurat FindMarkers function. The Student's t-test was used to calculate p-values. The p-values were adjusted with a Bonferroni correction using all genes in the dataset. Data exploration, analysis, and plotting were performed using RStudio (*RStudio Team, 2020*), the tidyverse collection of R packages (*Wickham, 2017*), and ggplot2 (*Wickham, 2009*).

## In situ hybridization

Eyes were enucleated and embedded in OCT on dry ice. Sections were cut at 18 μm thickness on a cryostat and mounted on Superfrost glass slides. ISH was performed using the RNAscope Fluorescent Multiplex Reagent kit (Advanced Cell Diagnostics 320850). The protocol was performed as recommended by the manufacturer, with minor modifications. Briefly, fresh frozen tissue sections were fixed with 4 % PFA in PBS at 4 °C for 15 min followed by dehydration in an ethanol series. Sections were treated with Protease III for 15 min. The following probes from Advanced Cell Diagnostics were used: Mm-Myh11-C1 (316101); Mm-Adra1a (408611-C2); Mm-Chrm3-C1 (437701); Mm-Chrm3-C2 (437701-C2); Mm-Nos1-C3 (437651-C3); Mm-Syt1-C2 (491831-C2); Mm-Ttn-C1 (483031); Mm-Slc26a4-C1 (452491); Mm-Tmem158-C3 (452721-C3); Mm-Btg2-C1 (483001); Mm-Anks1b-O1-C3 (577781-C3); Mm-Ntng1-C3 (488871-C3); Mm-Sntg1-C1 (440561); Mm-Pde10a-C1 (466201); and Mm-Junb-C1 (556651).

## Confocal microscopy

Confocal images were captured with a Zeiss LSM700 confocal microscope (40 × objective) using Zen Black 2012 software, and processed with ImageJ, Adobe Photoshop, and Adobe Illustrator software.

### Image analysis

Quantification of fluorescent RNAscope ISH signals (proportional to transcript abundances) was performed according to guidelines from Advanced Cell Diagnostics. Briefly, representative regions were selected on slides containing no RNAscope signal and the total integrated signal intensity was measured to provide an estimate of background intensity. The average background intensity was calculated as the integrated intensity divided by the area of the selected background region.

For each probe, 20 representative ISH signals (single 'dots') were chosen and the area and integrated intensity of each dot was determined to calculate the average intensity per dot with the following formula:

Average intensity per single dot = {(Total integrated intensity of selected dots) – [(Average background intensity)× Total area of selected dots]}/(Number of selected dots). The regions of interest (ROIs) were selected in the iris cross section slides, and the total ROI area was calculated and the total integrated intensity was measured. Using the average background intensity and the average intensity for each dot, the total number of dots in the ROI was estimated with the following formula: Total dot number in ROI = {(Total integrated intensity of ROI) – [(Average background intensity)× Total area of ROI]}/(Average intensity per single dot).

For each sample, the cross section of the iris was divided into two to three ROIs and the number of dots was estimated as described above for each ROI. To normalize for ISH efficiency in each section, the relative abundance of each transcript in each ROI was calculated as the ratio of the number of dots for that transcript divided by the number of dots for *Adra1a* (for dilator muscle analyses) or by the number of dots for *Chrm3* or *Syt1* (for sphincter muscle analyses).

### Quantifying nuclear morphology

Nuclear length and width were measured using Fiji-ImageJ (https://imagej.net/software/fiji/) from captured Z-stacked whole mount iris images. The longest dimension of each nucleus was measured and designated as the length, and the width was designated as the largest nuclear width when measured perpendicular to the length. For each condition and for each cell type, 150–500 nuclei were measured.

### Statistical analysis

All statistical values are presented as mean ± SD. Statistical tests were carried out using GraphPad Prism 8. Student's t-test was used to measure significance of differences between two independent groups. One-way analysis of variance (ANOVA) with appropriate multiple comparisons tests was used to compare three independent groups. The statistical significance is represented graphically as $*p < 0.05$; $**p < 0.01$; $***p < 0.001$; $****p < 0.0001$.

## Acknowledgements

This work was supported by the Howard Hughes Medical Institute. The authors thank the members of the Johns Hopkins Deep Sequencing Core for their assistance with snRNAseq, and Drs Jacob Heng, Xi Peng, and Yanshu Wang for helpful discussions and/or comments on the manuscript.

## Additional information

### Funding

| Funder | Grant reference number | Author |
|---|---|---|
| Howard Hughes Medical Institute | | Jeremy Nathans |

The funders had no role in study design, data collection and interpretation, or the decision to submit the work for publication.

### Author contributions

Jie Wang, Conceptualization, Formal analysis, Investigation, Methodology, Validation, Visualization, Writing - review and editing; Amir Rattner, Data curation, Formal analysis, Investigation, Validation, Visualization, Writing - review and editing; Jeremy Nathans, Conceptualization, Funding acquisition, Project administration, Supervision, Visualization, Writing - original draft, Writing - review and editing

### Author ORCIDs

Jie Wang http://orcid.org/0000-0002-6374-7043
Amir Rattner http://orcid.org/0000-0001-9542-6212
Jeremy Nathans http://orcid.org/0000-0001-8106-5460

### Ethics

All mice were housed and handled according to the approved Institutional Animal Care and Use Committee protocol of the Johns Hopkins Medical Institutions. (MO19M429).

### Decision letter and Author response

Decision letter https://doi.org/10.7554/eLife.73477.sa1
Author response https://doi.org/10.7554/eLife.73477.sa2

## Additional files

### Supplementary files

• Supplementary file 1. Single nucleus (sn)RNAseq library statistics.

• Supplementary file 2. The top 20 enriched transcripts for each iris and ciliary body cell type. FC, fold change (Excel File).

• Supplementary file 3. Transcripts showing the greatest fold change with dilation for each iris and ciliary body cell type. For each cell type, the table includes the 25 transcripts that showed the greatest fold increase and the 25 transcripts that showed the greatest fold decrease. FC, fold change (Excel File).

• Transparent reporting form

### Data availability

snRNAseq data has been deposited in the GEO database (NIH), accession numbers GSE183690 and GSM5567780-GSM5567787.

The following datasets were generated:

The following dataset was generated:

| Author(s) | Year | Dataset title | Dataset URL | Database and Identifier |
|---|---|---|---|---|
| Wang J, Rattner A, Nathans J | 2021 | A transcriptome atlas of the mouse iris at single cell resolution defines cell types and the genomic response to pupil dilation | https://www.ncbi.nlm.nih.gov/geo/query/acc.cgi?acc=GSE183690 | NCBI Gene Expression Omnibus, GSE183690 |
| Wang J, Rattner A, Nathans J | 2021 | snRNAseq_JW02_iris_control_RP1 | https://www.ncbi.nlm.nih.gov/geo/query/acc.cgi?acc=GSM5567780 | NCBI Gene Expression Omnibus, GSM5567780 |
| Wang J, Rattner A, Nathans J | 2021 | snRNAseq_JW03_iris_constricted_RP1 | https://www.ncbi.nlm.nih.gov/geo/query/acc.cgi?acc=GSM5567781 | NCBI Gene Expression Omnibus, GSM5567781 |
| Wang J, Rattner A, Nathans J | 2021 | snRNAseq_JW04_iris_dilated_RP1 | https://www.ncbi.nlm.nih.gov/geo/query/acc.cgi?acc=GSM5567782 | NCBI Gene Expression Omnibus, GSM5567782 |
| Wang J, Rattner A, Nathans J | 2021 | snRNAseq_JW13_iris_constricted_RP2 | https://www.ncbi.nlm.nih.gov/geo/query/acc.cgi?acc=GSM5567783 | NCBI Gene Expression Omnibus, GSM5567783 |
| Wang J, Rattner A, Nathans J | 2021 | snRNAseq_JW15_iris_constricted_RP3 | https://www.ncbi.nlm.nih.gov/geo/query/acc.cgi?acc=GSM5567784 | NCBI Gene Expression Omnibus, GSM5567784 |
| Wang J, Rattner A, Nathans J | 2021 | snRNAseq_JW16_iris_dilated_RP2 | https://www.ncbi.nlm.nih.gov/geo/query/acc.cgi?acc=GSM5567785 | NCBI Gene Expression Omnibus, GSM5567785 |

*Continued on next page*

*Continued*

| Author(s) | Year | Dataset title | Dataset URL | Database and Identifier |
|---|---|---|---|---|
| Wang J, Rattner A, Nathans J | 2021 | snRNAseq_JW17_iris_control_RP2 | https://www.ncbi.nlm.nih.gov/geo/query/acc.cgi?acc=GSM5567786 | NCBI Gene Expression Omnibus, GSM5567786 |
| Wang J, Rattner A, Nathans J | 2021 | snRNAseq_JW18_iris_control_RP3 | https://www.ncbi.nlm.nih.gov/geo/query/acc.cgi?acc=GSM5567787 | NCBI Gene Expression Omnibus, GSM5567787 |

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
