## [Editor Report]

Using single nucleus RNA sequencing, the authors have characterized all major cell types in the mouse iris and ciliary body, defined new types of iris stromal and sphincter cells, and shown cell-specific transcriptome responses in the resting, constricted, and dilated states. They have identified and validated antibodies and in situ hybridization probes for visualization of major iris cell types. This work will be a valuable reference for investigations of iris development, disease, and pharmacology.

---

## [Decision Letter]

**Decision letter after peer review:**

Thank you for submitting your article "A transcriptome atlas of the mouse iris at single cell resolution defines cell types and the genomic response to pupil dilation" for consideration by *eLife*. Your article has been reviewed by 3 peer reviewers, including Lois Smith as the Reviewing Editor and Reviewer #1, and the evaluation has been overseen by Marianne Bronner as the Senior Editor. The following individuals involved in review of your submission have agreed to reveal their identity: Bo Chen (Reviewer #2); Chenxi Qiu (Reviewer #3).

Essential revisions:

1) 10x platform was used in this study. The authors need to include: 1) If all the samples were collected in a single round of experiment on the same day. If not, how the batch effect was minimized. 2) For the three groups (untreated, dilated, constricted) of samples, what steps were used to minimize the potential transcriptional change and variation during the sample preparation. 3) How long after dilation or constriction were the sample taken?

2) Although only 6-8-week old female mice were used for single cell analysis, were the validation also conducted in male mice?

3) Does the duration of pupil dilation and constriction affect the transcriptome analysis? How did the authors choose seven hours as the duration for the dilation and constriction, any preparatory analysis to determine this time point?

4) Some missing labeling in the text, including Figure 2A, Figure 3A, and Figure 7A.

5) Could the authors provide some thoughts on why only dilation has effects on the gene expression changes, but not constriction? Some discussion on this might be helpful. No experimental proof required.

6) The methodological details for the differential gene expression analyses are not discussed in the manuscript. In addition, the authors should provide the full list of differentially expressed genes instead of top 25 genes and discuss the cutoff of differentially expressed genes where they are confident. The adjusted p-values for the top differentially expressed genes are small and may not provide true value into the analyses.

7) The figure legends could have more details on the replicates shown on the quantitation of immunofluorescence images. Example1 : In figure 8, I am assuming each dot here represents one nucleus. How many mice are in this plot? Is there systematic variation among different mice? It would help if the authors color code the data points by the mouse each nucleus is from. Example2: In supplemental figure 4, is each data point here is from one mouse or one image for a given area, and how many mice were used.

8) The choice of snRNA-seq over single-cell RNA-seq is well justified, as uniform sampling of diverse cell types is more critical for defining major cell types. However, the authors should discuss limitations of snRNA-seq in case future studies are built upon this work. For example, in cases where cells are enzymatically digestible and cytoplasmic mRNAs are of interests, scRNA-seq is a more relevant approach. It's also worth noting that the snRNA-seq approach in this work could still provide a roadmap for future scRNA-seq analyses.

9) The library preparation method should be expanded. The authors cited the 10X Genomics Chromium single cell 3' v3 kit, which may not be sufficient. Different RNA species are largely sensitive to the library preparation methods, especially for snRNA-seq. This is an essential section in the methods section that needs to be expanded.

10) In the differential gene expression analyses, is each cell treated as a replicate? If so how did the authors account for the biological replicates from different mice? How did the authors account for the dissection variation among different mice? Was there any quality controls done among different mice in the same group in terms of dissection variation?

11) Dotplots in the supplemental figure 3 lacks statistical tests and could be sometimes misleading, as the color of each dot is Z-score normalized if performed using the default setting of Seurat and could unnecessarily exaggerate an effect. The readers will benefit from some indication of adjusted p-values on the figure (e.g. asterisks). However, the authors should first clarify how different gene expression analyses were done. It would be helpful to show how a small effect size as validated in supplemental figure 4 could have so much statistical confidence as shown in the supplemental table 3.

*Reviewer #1:*

This study establishes fundamental information on the mouse iris and its function. Using single nucleus RNA sequencing, the authors have characterized all major cell types in the mouse iris and ciliary body, defined two types of iris stromal cells and two types of iris sphincter cells, and shown cell-specific transcriptome responses in the resting, constricted, and dilated states. They have identified and validated antibody and in situ hybridization probes for visualization of the major iris cell types. They have quantified distortions in nuclear morphology associated with iris dilation and clarified the neural crest contribution to the iris by showing that Wnt1-Cre-expressing progenitors contribute to nearly all iris cell types, whereas Sox10-Cre expressing progenitors contribute only to stromal cells. This work will be a valuable reference for investigations of iris development, disease, and pharmacology, for the isolation and propagation of defined iris cell types, and for iris cell engineering and transplantation.

This paper was a pleasure to read. It is well written, thorough, and will provide tools to study the iris and ciliary body for the research community. I had no major concerns.

*Reviewer #2:*

Major strengths of the manuscript:

1) Using single nucleus RNA sequencing technology had several advantages over single cell RNA sequencing with minimum disturbance of the native transcriptional profiles.

2) This research revealed major cell types in the mouse iris and provided valuable and verifiable markers for each of the iris cell types. This research generated great resources for future studies on normal and diseased irises.

3) The study showed very interesting changes in the transcriptome and nuclear morphology associated with iris dilation, and the most upregulated genes identified could be great candidates for studying iris function and malfunction in diseases.

4) The study provided definitive experimental proof showing the neural crest contribution to the various iris cell types.

Overall, the study was well designed and precisely executed, the data analysis was clear and scientifically stringent, the results are comprehensive and revealing novel molecular correlates of cellular responses.

*Reviewer #3:*

This work defines the mouse iris transcriptomic atlas by single-nucleus RNA-seq (snRNA-seq), an approach that captures nuclear transcripts without enzymatic cell dissociation and processing. The major cell types defined/revealed are independently and rigorously validated by immunofluorescence and fluorescence in-situ hybridization. Immunofluorescence and fluorescence in-situ hybridization experiments further confirmed distinction between sphincter and dilator muscles and revealed distinct distribution of subtypes of sphincter and stromal cells. More importantly, the snRNA-seq approach they have undertaken, though only capturing the nuclear transcripts, is sufficient to profile the transcriptomic changes during constriction and dilation, and some of the expression changes were confirmed by immunofluorescence. The identification of transcription factors associated with defined cell types also allows tests of an unexplored question- does nuclear morphology change along with known changes in the cell plasma during dilation? The authors assessed the nuclear morphology of each cell type by immunofluorescence of cell-type specific transcription factors they identified from snRNA-seq in this study, and found cell-type specific changes of nuclear morphology during dilation. Finally, the authors revisited a partially conflicting result on the neural crest cells contribution to iris cell types, with characterized transcription factors in this study to increase resolution.

Overall, this is a rigorous study and could have broad interests. This version of manuscript could benefit from more details in statistics and methodology in some analyses. Despite the insufficient technical/statistical details in some figures, the authors' major claims and the identified sub-celltypes are justified by their data.

---

## [Author Response]

Essential revisions:1) 10x platform was used in this study. The authors need to include:1) if all the samples were collected in a single round of experiment on the same day. If not, how the batch effect was minimized.

We have added the following text to the Methods section:

“The eight groups of irises were collected and processed with the 10X Genomics protocol in 5 batches, and the libraries were sequenced in two runs. The different conditions (constricted, dilated, and untreated) were each distributed across two or more batches and libraries from each condition were present in each of the two sequencing runs.”

Regarding batch effects, we have added the following to the Methods section:

“After the data was normalized using a regularized negative binomial regression algorithm (implemented in the SCTransform function as described in Hafemaister and Satija (2019)), there appeared to be little or no batch effects as judged by the high pairwise Pearson correlations (0.98-0.99 within each condition) and the highly similar UMAPs for each sample (see the new Figure 1—figure supplements 1 and 2)”.

2) For the three groups (untreated, dilated, constricted) of samples, what steps were used to minimize the potential transcriptional change and variation during the sample preparation.

The irises were dissected and rapidly homogenized in ice-cold buffer for nuclear isolation. We assume that the tissue disruption and the high dilution of small molecules (rNTPs, etc) efficiently inhibits RNA synthesis and degradation.

3) How long after dilation or constriction were the sample taken?

As described in the Methods section, irises in living mice were maintained in the dilated or constricted states by hourly instillation of eyedrops with the appropriate drugs. Thirty minutes after the last eyedrop treatment the mice were sacrificed and the irises harvested. This is now made clearer in the Results by the phrases “continuous dilation” and “continuous constriction”.

2) Although only 6-8-week old female mice were used for single cell analysis, were the validation also conducted in male mice?

No, we have not studied male mice. We decided to use only female mice to minimize potential variation associated with sex, and because female mice can be housed together without the fighting and social stress that characterizes group housing of males.

3) Does the duration of pupil dilation and constriction affect the transcriptome analysis?

Thank you for raising this question, which have also been wondering about. For the revised manuscript we performed a time course of dilation (30 minutes, 90 minutes, and 150 minutes) and analyzed the response by immunostaining for EGR1. Interestingly, EGR1 induction is already present at 30 minutes. This has now been added to Figure 7 as a new panel D. Since there were no transcriptome changes with 6.5 hours of constriction, we did not explore shorter constriction times.

How did the authors choose seven hours as the duration for the dilation and constriction, any preparatory analysis to determine this time point?

Six and one-half hours was chosen as the duration for constriction and dilation because we hypothesized that this would provide a sufficiently long time to allow for changes in gene expression to be manifest. This estimate was based on other physiologic responses in mammals and mammalian cells in culture, such as responses to growth factors, metabolic perturbations, etc, which are generally maximal within several hours of onset. This is now clarified in the Results section with the sentence “Six and one-half hours was chosen as the duration for dilation or constriction to provide sufficient time for any changes in gene expression to be fully manifest.”

4) Some missing labeling in the text, including Figure 2A, Figure 3A, and Figure 7A.

Thank you for noting this. In each of these instances, we had referred to the entire figure. In the revised manuscript, we now refer to each panel.

5) Could the authors provide some thoughts on why only dilation has effects on the gene expression changes, but not constriction? Some discussion on this might be helpful. No experimental proof required.

The iris is like an accordion or a set of pleated window shades – when the pupil dilates, the iris tissue is compressed, with blood vessels and nerves zig-zagging back and forth (Figure 6B (formerly 6A)), nuclei flattening (Figure 8), and plasma membranes of the epithelia folding to create a bumpy surface (Lim and Webber, 1975). By contrast, constriction flattens out the iris. The untreated iris is in-between these extremes (see the new images in Figure 6A). Our guess is that the gene expression changes with dilation reflect the cell biologic distortions associated with tissue compression. We have added the following text to the last section of the Discussion:

“The asymmetry in gene expression responses, in which an iris with a dilated pupil differs from an untreated iris but an iris with a constricted pupil is nearly identical to an untreated iris, most likely reflects the tissue compression associated with pupil dilation, as seen in the zig-zagging morphology of blood vessels and nerves (Figure 6B), the flattening of nuclear shape (Figure 8), and the membrane infoldings demonstrated in ultrastructural studies (Lim and Webber, 1975a, 1975b). By contrast, with pupil constriction, the iris is flattened to an even greater extent than it is in the untreated (resting) state.”

6) The methodological details for the differential gene expression analyses are not discussed in the manuscript. In addition, the authors should provide the full list of differentially expressed genes instead of top 25 genes and discuss the cutoff of differentially expressed genes where they are confident. The adjusted p-values for the top differentially expressed genes are small and may not provide true value into the analyses.

Thank you for that comment. In the revised manuscript, we have expanded the Methods section “Analysis of snRNAseq data” to flesh this out. We have also changed our analysis of differential transcripts between dilated and untreated irises by applying a defined threshold (fold change greater than log2=0.6 and P-value less than 0.05). These threshold criteria are illustrated in the new Figure 6—figure supplement 2 and applied to the updated Figure 6—figure supplement 1 and Supplemental Table 3. For the endothelial cell population, no transcripts passed this threshold, and therefore Figure 6—figure supplements 1 and 2 and Supplemental Table 3 list only the other nine cell types. Re: adjusted P-values. Because snRNAseq and scRNAseq calculate read-count statistics with each nucleus or cell as an independent data point and because the evaluation is based on a data set with several hundred million reads, even small fold-changes tend to be associated with very small adjusted Pvalues (Bonferroni corrected). Depending how you look at it, this is either a strength or a weakness of single cell statistical analysis. From a purely statistical point of view, it is a strength because this is an objective analysis of the data. On the other hand, from the point of view of the reader trying to get an intuitive grasp of what the Pvalue means, it requires a recalibration from our customary familiarity with small sample sizes.

7) The figure legends could have more details on the replicates shown on the quantitation of immunofluorescence images. Example1 : In figure 8, I am assuming each dot here represents one nucleus. How many mice are in this plot? Is there systematic variation among different mice? It would help if the authors color code the data points by the mouse each nucleus is from. Example2: In supplemental figure 4, is each data point here is from one mouse or one image for a given area, and how many mice were used.

Thank you for that comment. We have expanded the legend of Figure 2 (the first figure to show immunostaining) with the following sentence:

“For each immunostaining analysis in this and subsequent figures, iris cross-sections were stained from at least five mice and iris whole mounts were stained from at least two mice.”

We have expanded the legend of Figure 3 (the first figure to show in situ hybridization) with the following sentence:

“For each in situ hybridization analysis in this and subsequent figures, iris cross-sections were hybridized from at least three mice.”

In figure 8, each dot represents one nucleus. To assess systematic variation between mice, we have plotted the data for individual mice in a new Figure 8—figure supplement 1, which shows that the distribution of nuclear length/width ratios for individual mice are consistent within each of the three conditions (constricted, dilated, and untreated) and are also consistent with the aggregate data presented in Figure 8. In Figure 7—figure supplement 1 (formerly named Supplemental Figure 4), we now explain in the figure legend “Each data point is derived from a single image and represents approximately one-quarter of the iris cross-sectional area from pupil to periphery. For each condition and probe, 3-4 mice were analyzed.”

8) The choice of snRNA-seq over single-cell RNA-seq is well justified, as uniform sampling of diverse cell types is more critical for defining major cell types. However, the authors should discuss limitations of snRNA-seq in case future studies are built upon this work. For example, in cases where cells are enzymatically digestible and cytoplasmic mRNAs are of interests, scRNA-seq is a more relevant approach. It's also worth noting that the snRNA-seq approach in this work could still provide a roadmap for future scRNA-seq analyses.

Thank you for that comment – we agree. We have added the following statement to the beginning of the Results section:

“A potential disadvantage of snRNAseq relative to scRNAseq is that it is insensitive to changes in RNA abundance arising from changes in RNA stability in the cytoplasm.”

9) The library preparation method should be expanded. The authors cited the 10X Genomics Chromium single cell 3' v3 kit, which may not be sufficient. Different RNA species are largely sensitive to the library preparation methods, especially for snRNA-seq. This is an essential section in the methods section that needs to be expanded.

We followed the 10X genomics V3 protocol, which is unfortunately not published in a peer-reviewed journal, but is available as a pdf download from the 10X Genomics web site. We now cite that web site and make it clear that we are following that protocol. “snRNAseq libraries were constructed using the 10X Genomics Chromium single cell 3’ v3 kit and following the manufacturer’s protocol (https://support.10xgenomics.com/single-cellgene-expression/library-prep/doc/user-guide-chromium-single-cell-3-reagent-kits-user-guide-v31-chemistry).” This non-transparency issue permeates the field since virtually everyone is using kits – many with proprietary reagents – for Nextgen sample analyses.

10) In the differential gene expression analyses, is each cell treated as a replicate?

Yes, each cell’s RNAseq read counts are an independent data point.

If so how did the authors account for the biological replicates from different mice?

We have added the following statement to the Results section and added Figure 1—figure supplements 1 and 2 to illustrate the point:

“Among independent replicates from mice subject to the same treatment, RNAseq read counts showed pairwise Pearson correlations of 0.98-0.99 (Figure 1—figure supplement 1). Therefore, for all subsequent analyses, each set of replicate samples were merged into a single data set.”

How did the authors account for the dissection variation among different mice?

We have added the following to the methods section:

“Subsequent snRNAseq nucleus counts show that the percent of cells derived from CB and CBE to be tightly correlated within each sample (R^2^=0.9) and together they vary from 3% to 15% of the total cell number – presumably reflecting variable inclusion of the ciliary body as a result of variability in cutting around the margin of the iris. Nucleus counts for other cell types, which are intrinsic to the iris, showed less variation between samples.”

Was there any quality controls done among different mice in the same group in terms of dissection variation?

No, we simply pooled 5-6 irises per snRNAseq run and figured that the variation in cutting around the margins of the iris would average out. This turned out to be roughly correct, as judged by the inclusion in every data set of ciliary body tissue – which resides just beyond the iris.

11) Dotplots in the supplemental figure 3 lacks statistical tests and could be sometimes misleading, as the color of each dot is Z-score normalized if performed using the default setting of Seurat and could unnecessarily exaggerate an effect. The readers will benefit from some indication of adjusted p-values on the figure (e.g. asterisks). However, the authors should first clarify how different gene expression analyses were done. It would be helpful to show how a small effect size as validated in supplemental figure 4 could have so much statistical confidence as shown in the supplemental table 3.

We have fleshed out this issue by producing a new supplemental figure (Figure 6—figure supplement 2) that shows volcano plots with a label for each transcript shown in the dot plot in supplemental figure 3 (now re-named Figure 6—figure supplement 1). As discussed in the reply to comment 6 above, the statistical analysis of scRNAseq takes some recalibrating on the part of the reader. “Because snRNAseq and scRNAseq calculate read-count statistics with each nucleus or cell as an independent data point and because the evaluation is based on a data set with several hundred million reads, even small fold-changes tend to be associated with very small adjusted P-values (Bonferroni corrected). Depending how you look at it, this is either a strength or a weakness of single cell statistical analysis. From a purely statistical point of view, it is a strength because this is an objective analysis of the data. On the other hand, from the point of view of the reader trying to get an intuitive grasp of what the P-value means, it requires a recalibration from our customary familiarity with small sample sizes.”